# Equivariant Motion Manifold Primitives

**Byeongho Lee**[*1]    **Yonghyeon Lee**[*2]    **Seungyeon Kim**[1]    **Minjun Son**[1]    **Frank C. Park**[1]

[1] Seoul National University    [2] Korea Institute for Advanced Study (KIAS)
{bhlee, ksy, mjson}@robotics.snu.ac.kr    ylee@kias.re.kr    fcp@snu.ac.kr

**Abstract:** Existing movement primitive models for the most part focus on representing and generating a single trajectory for a given task, limiting their adaptability to situations in which unforeseen obstacles or new constraints may arise. In this work we propose Motion Manifold Primitives (MMP), a movement primitive paradigm that encodes and generates, for a given task, a continuous manifold of trajectories each of which can achieve the given task. To address the challenge of learning each motion manifold from a limited amount of data, we exploit inherent symmetries in the robot task by constructing motion manifold primitives that are equivariant with respect to given symmetry groups. Under the assumption that each of the MMPs can be smoothly deformed into each other, an autoencoder framework is developed to encode the MMPs and also generate solution trajectories. Experiments involving synthetic and real-robot examples demonstrate that our method outperforms existing manifold primitive methods by significant margins. Code is available at https://github.com/dlsfldl/EMMP-public.

**Keywords:** Movement primitives, Manifold, LfD, Equivariance

## 1 Introduction

Learning basic motion skills as movement primitives has been an enduring focus of learning from demonstration (LfD) research [1, 2, 3]. A primary challenge is constructing movement primitive models adaptable to diverse situations, such as when unforeseen obstacles or new constraints emerge. Current approaches to movement primitives encompass dynamic movement primitives [4, 5, 6, 7, 8, 9, 10, 11, 12, 13], stable dynamical systems [14, 15, 16, 17, 18, 19, 20, 21, 22], methods based on Gaussian processes [23, 24, 25] and Gaussian mixture models [26, 27, 28, 29], along with other methods [30, 31, 32].

The limited adaptability of existing primitive models largely stems from their design which encodes and generates a single trajectory for a specific task, since they have no alternatives when their primary trajectory becomes infeasible in new environments (e.g. when an unexpected obstacle blocks the trajectory). Although dynamical system-based methods can integrate, for instance, obstacle avoidance potential function terms [33, 34, 35, 36], the resulting motions might violate other task constraints. For adaptable motion primitives, a method that encodes multiple trajectories for a single task is essential.

In this paper, we propose to learn a continuous manifold of motion trajectories that can perform the given task, which we refer to as Motion Manifold Primitives (MMP). As the MMP encodes multiple successful trajectories, even if some trajectories are obstructed by obstacles or violate constraints, alternative feasible trajectories remain accessible within the MMP. As such, the MMP is highly adaptable, although more diverse demonstration data are needed for training, compared to the single trajectory-based primitives.

Given a set of task-trajectory paired data – where multiple demonstration trajectories are collected for a single task parameter $\tau$ –, our objective is to learn a set of manifold primitives $\{\mathcal{M}_\tau\}$ for

---

[*] The two lead co-authors contributed equally.

7th Conference on Robot Learning (CoRL 2023), Atlanta, USA.

all $\tau$. However, given a limited amount of demonstration data, learning accurate manifolds and their boundaries is a very challenging task. In fact, the TC-VAE [37] is the first work to adopt this manifold primitives approach but shows less-than-desirable performance given a small dataset as we show later in our experiments.

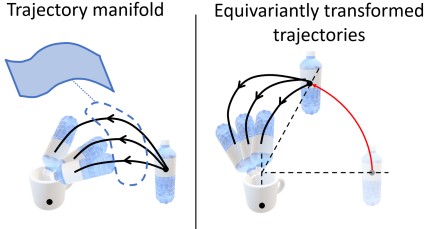

In this paper, we develop a data-efficient motion manifold primitives learning algorithm, where we adopt the autoencoder-based manifold learning framework [38, 39, 40, 41, 42, 43, 44]. First, we propose *Equivariant Motion Manifold Primitives (EMMP)*, which takes into account the inherent symmetry in robot tasks. For example, consider a water-pouring task where the initial cup and bottle's positions are defined as the task parameter. Given a symmetry transformation on $\tau$ that preserves the relative positional relation between the cup and bottle (e.g., rotating bottle around the cup), the set of water-pouring trajectories, the motion manifold primitives

Figure 1: An illustration of the motion manifold primitives and equivariant transformation.

$\mathcal{M}_\tau$, should also be transformed in a consistent manner, more precisely, *equivariantly* (see Figure 1). We show that using an invariant encoder and equivariant decoder can guarantee the equivariance of the MMP and propose a strategy to construct invariant and equivariant mappings. Meanwhile, equivariance has been increasingly recognized as a pivotal factor in robotics tasks in general [45, 46, 47].

Second, to further enhance data efficiency, we find a shared latent coordinate space $\mathcal{Z}$, by assuming that $\mathcal{M}_\tau$ for all $\tau$ are homeomorphic (i.e., can be smoothly deformed into each other). In practice, we consider the latent coordinate variable $z \in \mathcal{Z}$ to be independent of $\tau$. This means that $p(z|\tau) = p(z)$ for all $\tau$, making the support of $p(z)$ the shared latent coordinate space $\mathcal{Z}$. To enforce this condition, we newly propose an *independence regularization term* in autoencoder training.

Through comprehensive experiments involving synthetic and real-world robot water-pouring experiments, we compare our EMMP model with the existing manifold primitives method TC-VAE [37] through a systematic evaluation on (i) manifold learning, (ii) independence between $z$ and $\tau$, (iii) density learning, and (iv) success rate. We find that our method significantly outperforms TC-VAE. Further, an in-depth ablation study reveals that the use of an invariant encoder and equivariant decoder is the primary factor driving this performance improvement.

## 2 MMPs: Motion Manifold Primitives

In this section, given a configuration space $\mathcal{Q}$, we consider a motion as a discrete and fixed-length configuration trajectory $x = \{q_t \in \mathcal{Q}\}_{t=1}^T$; the space of all trajectories is denoted by $\mathcal{X} := \mathcal{Q}^T$. Let a compact space $\mathcal{T}$ represent a

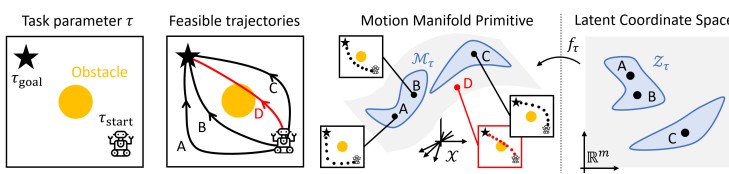

Figure 2: An illustration of a task parameter $\tau$, feasible trajectories that achieve the task, and a manifold that the trajectories form. The resulting manifold $\mathcal{M}_\tau$ consists of two disjoint components.

space of parameters that define specific tasks. We assume that, for each $\tau \in \mathcal{T}$, a set of motions that can perform the given task forms an $m$-dimensional differentiable manifold, which we refer to as the Motion Manifold Primitives (MMP) denoted by $\mathcal{M}_\tau$, and that is a submanifold lying in the ambient space $\mathcal{X}$. We assume that we are given a set of task-trajectory paired data, denoted by $\mathcal{D} := \{(\tau_i, x_{ij}) \in \mathcal{T} \times \mathcal{X}\}_{i=1,\dots,N, j=1,\dots,M_i}$, where $M_i$ represents the total number of trajectories for the task parameter $\tau_i$ and $x_{ij}$ is sampled from $\mathcal{M}_{\tau_i}$. For readers unfamiliar with geometric concepts such as manifold and homeomorphism, we include brief introductions of them in Appendix C.

## 2.1 MMP described via Manifold and Density

This section introduces how to represent the set of motion manifold primitives $\mathcal{M}_\tau \subset \mathcal{X}$ for all $\tau \in \mathcal{T}$. First, we assume that each manifold primitive $\mathcal{M}_\tau$ can be parametrized by a nonlinear map $f_\tau : \mathbb{R}^m \to \mathcal{X}$ with a coordinate space $\mathcal{Z}_\tau \subset \mathbb{R}^m$ as $\mathcal{M}_\tau = f_\tau(\mathcal{Z}_\tau)$. To represent the set of motion manifold primitives simultaneously, we consider a differentiable mapping $f : \mathbb{R}^m \times \mathcal{T} \to \mathcal{X}$ such that $f(z, \tau) := f_\tau(z)$. Second, to represent each coordinate space $\mathcal{Z}_\tau$, we employ a conditional probability density function in $\mathbb{R}^m$ given $\tau$ denoted by $p(z|\tau)$. Then, we consider the support of $p(z|\tau)$ as $\mathcal{Z}_\tau$. As a result, each motion manifold for $\tau$ is represented as $\mathcal{M}_\tau = f(\mathcal{Z}_\tau, \tau)$; the motion manifold primitives $\mathcal{M}_\tau$ is said to be parametrized by the mapping $f$ and the density $p(z|\tau)$ or parametrized by $f$ and $\mathcal{Z}_\tau$ (see Figure 2).

## 2.2 Autoencoder-based Manifold and Density Learning

We adopt the autoencoder framework for learning $\mathcal{M}_\tau$ via $f(z, \tau)$ and $p(z|\tau)$, where $f$ is considered as the decoder and $p$ as the latent space density in $\mathbb{R}^m$. An additional component that specifies the coordinates of an input trajectory $x \in \mathcal{M}_\tau$ needs to be introduced, called an encoder, and we denote it by $g : \mathcal{X} \times \mathcal{T} \to \mathbb{R}^m$ such that $z = g(x, \tau)$. We use parametric models (e.g., neural networks) for encoder $g_\phi$, decoder $f_\theta$, and density $p_\gamma(z|\tau)$, where $\theta$, $\phi$, and $\gamma$ represent the model parameters. We propose a two-step approach where we first learn the coordinate systems, i.e., $f_\theta, g_\phi$, and then learn the density, i.e., $p_\gamma(z|\tau)$.

The standard autoencoder reconstruction loss can be employed to learn $f_\theta, g_\phi$, that is an expectation of $d_\mathcal{X}^2(f_\theta(g_\phi(x_{ij}, \tau_i), \tau_i), x_{ij})$ where $d_\mathcal{X}(\cdot, \cdot)$ is a proper distance measure on $\mathcal{X}$. Minimizing the reconstruction loss results in that $f_\theta(g_\phi(\mathcal{M}_\tau, \tau), \tau) \approx \mathcal{M}_\tau$ for the ground truth manifold $\mathcal{M}_\tau$. $f_\theta(\cdot, \tau)$ becomes a coordinate system for $\mathcal{M}_\tau$ and $g_\phi(\mathcal{M}_\tau, \tau)$ becomes the coordinate space $\mathcal{Z}_\tau$. Secondly, we can learn $p_\gamma(z|\tau)$ given a trained encoder $g_\phi$ via standard likelihood maximization framework, i.e., maximizing an expectation of $\log p_\gamma(g_\phi(x_{ij}, \tau_i)|\tau_i)$.

## 2.3 Homeomorphic Manifold Assumption

Learning densities for all $\tau$, i.e., $p(z|\tau)$ is challenging given the small training dataset $\mathcal{D}$. In this section, we introduce a homeomorphic manifold assumption to make the density learning problem more tractable. We assume that $\mathcal{M}_\tau$ for each $\tau$ are homeomorphic to each other and there exists a latent coordinate variable $z$ statistically independent of $\tau$, i.e. $p(z) = p(z|\tau)$ for all $\tau$, which results in a shared latent coordinate space $\mathcal{Z} = \mathcal{Z}_\tau$ for all $\tau$. The training of autoencoder and density can be greatly simplified under this assumption.

First, we can restrict the encoder's input space to $\mathcal{X}$, i.e., $g : \mathcal{X} \to \mathbb{R}^m$, because $\tau$ should not contribute to $z$. Second, $p(z|\tau)$ can be replaced by a shared model $p(z)$. Using $g(x)$ and $p(z)$, we may sequentially train the autoencoder and density, however, the reconstruction loss alone does not guarantee the statistical independence between $z$ and $\tau$, hence learning the density $p(z)$ can fail.

In this section, we introduce an independence regularization term for autoencoder training, so that $z$ and $\tau$ become independent and $\mathcal{Z} = \mathcal{Z}_\tau$ for all $\tau$:

$$\mathcal{R}(\theta, \phi) := \frac{\mathbb{E}_{(\cdot, x_{ij}) \in \mathcal{D}}\left[\|g_\phi(x_{ij}) - g_\phi(f_\theta(g_\phi(x_{ij}), \tau))\|^2\right]}{\mathbb{E}_{(\cdot, x_{ij}) \in \mathcal{D}}\|g_\phi(x_{ij})\|^2}, \tag{1}$$

where $\tau$ is randomly sampled from the uniform distribution on $\mathcal{T}$. This loss enforces that $g(f(z, \tau))$ does not depend on $\tau$, and eventually, $\tau$ does not contribute to $z$, and becomes independent of $z$. The denominator makes the regularization term invariant to the latent value scale. This regularization term $\mathcal{R}$ is added to the reconstruction loss with a proper regularization coefficient.

# 3 EMMPs: Equivariant Motion Manifold Primitives

In this section, our goal is to construct the motion manifold primitive $\mathcal{M}_\tau$ that transforms equivariantly when symmetry transformations are applied to $\tau$. We denote a symmetry group by $H$ where

the group operation between two elements $h_1, h_2 \in H$ is written as $h_1 h_2 \in H$. Then we assume symmetry transformations are defined as group actions. Two symmetry transformations in $\mathcal{T}$ and $\mathcal{X} \times \mathcal{T}$ are considered, where we use the same symbol to denote the group action in $\mathcal{T}$ by $\tau \mapsto h \cdot \tau$ and that in $\mathcal{X} \times \mathcal{T}$ by $(x, \tau) \mapsto h \cdot (x, \tau)$. Let $\big[ \cdot \big]_x$ denote the $x$ component, i.e., $\big[(x, \tau)\big]_x = x$. The ground truth motion manifold primitive is denoted by $\mathcal{M}_\tau$, while the learned motion manifold primitive is denoted by $\widehat{\mathcal{M}}_\tau$ parametrized by a decoder $f(z, \tau)$ and $p(z|\tau)$. We include preliminary knowledge on group, equivariance, and invariance in Appendix C.

## 3.1 Invariant Encoder and Equivariant Decoder

We begin this section with the definition of the equivariant motion manifold primitive:

**Definition 3.1.** Suppose $\mathcal{M}_\tau$ is a motion manifold primitive. Given a transformed task parameter $\tau \mapsto h \cdot \tau$, if the motion manifold $\mathcal{M}_\tau$ is equivariantly transformed, i.e., $\mathcal{M}_{h \cdot \tau} = \big\{ \big[ h \cdot (x, \tau) \big]_x \mid x \in \mathcal{M}_\tau \big\}$, then the primitive is called an Equivariant Motion Manifold Primitive (EMMP); see Figure 3.

We show that using an invariant encoder and an equivariant decoder can guarantee the equivariance of the resulting MMP; we first provide their definitions:

**Definition 3.2.** An encoder $g : \mathcal{X} \times \mathcal{T} \to \mathbb{R}^m$ is invariant if $g(h \cdot (x, \tau)) = g(x, \tau)$ for all $h \in H$ and $x \in \mathcal{X}$.

**Definition 3.3.** A decoder $f : \mathbb{R}^m \times \mathcal{T} \to \mathcal{X}$ is equivariant if $f(z, h \cdot \tau) = \big[ h \cdot (f(z, \tau), \tau) \big]_x$ for all $h \in H$ and $\tau \in \mathcal{T}$.

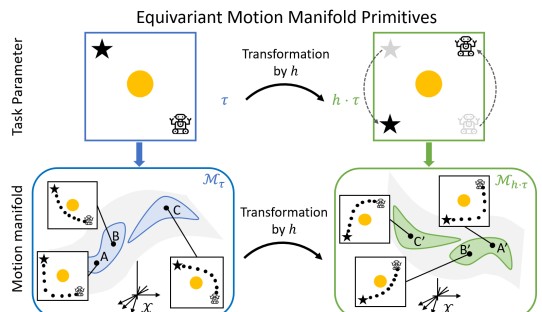

Equivariant Motion Manifold Primitives

Figure 3: Given a symmetry transformation to the task parameter $\tau \mapsto h \cdot \tau$ where the relative distances between the robot, goal, and obstacle are preserved, the MMP is equivariantly transformed.

Denoting the ground truth motion manifold by $\mathcal{M}_\tau$, the latent coordinate space $\mathcal{Z}_\tau$ defined as the support of $p(z|\tau)$ can be specified by an encoder $g(x, \tau)$ as follows: $\mathcal{Z}_\tau = \left( \bigcup_{x \in \mathcal{M}_\tau} g(x, \tau) \right)$ or $g(\mathcal{M}_\tau, \tau)$. If $\mathcal{M}_\tau$ is equivariant, then an invariant encoder produces invariant coordinate space:

**Proposition 3.1.** *Suppose $\mathcal{M}_\tau$ is an EMMP. If $g$ is invariant, then $\mathcal{Z}_\tau = \mathcal{Z}_{h \cdot \tau}$ for all $h \in H$.*

In addition to the encoder invariance condition, the learned MMP parametrized by a decoder $f(z, \tau)$ and $\mathcal{Z}_\tau$, denoted by $\widehat{\mathcal{M}}_\tau$, is equivariant if a decoder $f$ is equivariant:

**Proposition 3.2.** *Suppose $\mathcal{M}_\tau$ is an EMMP. If $g$ is invariant and $f$ is equivariant, then the MMP parametrized by $f$ and $\mathcal{Z}_\tau$, i.e., $\widehat{\mathcal{M}}_\tau = f(\mathcal{Z}_\tau, \tau)$ where $\mathcal{Z}_\tau = g(\mathcal{M}_\tau, \tau)$, is equivariant.*

By constructing an invariant encoder and an equivariant decoder, we can ensure the equivariance of the learned motion manifold primitives. When the ground truth MMP is equivariant, it is reasonable to expect that this equivariance guarantee would enhance the accuracy of manifold learning.

## 3.2 Construction of Invariant and Equivariant Mappings

In this section, we propose a method for converting arbitrary encoder and decoder models to ones that are invariant and equivariant to the symmetry group $H$. Let $G_\phi : \mathcal{X} \times \mathcal{T} \to \mathbb{R}^m$ and $F_\theta : \mathbb{R}^m \times \mathcal{T} \to \mathcal{X}$ be arbitrary parametric models for encoder and decoder, respectively. Assuming we can find an equivariant map $\bar{h} : \mathcal{T} \to H$ such that $\bar{h}(h \cdot \tau) = h \bar{h}(\tau)$ for all $h \in H, \tau \in \mathcal{T}$, an invariant encoder and equivariant decoder can be constructed as follows:

**Proposition 3.3.** *An encoder $g : \mathcal{X} \times \mathcal{T} \to \mathbb{R}^m$ defined as $g_\phi(x, \tau) := G_\phi(\bar{h}(\tau)^{-1} \cdot (x, \tau))$ is invariant and a decoder $f : \mathbb{R}^m \times \mathcal{T} \to \mathcal{X}$ defined as $f_\theta(z, \tau) := \big[ \bar{h}(\tau) \cdot (F_\theta(z, \bar{h}(\tau)^{-1} \cdot \tau), \bar{h}(\tau)^{-1} \cdot \tau) \big]_x$ is equivariant.*

How to construct the map $\bar{h}$ is problem-specific. As an example, let $\mathcal{X}$ be the set of 3D point cloud data – a point $x \in \mathcal{X}$ is of the form $x = \{x_1, \ldots, x_N\}$ – and $H$ be the group of translations in $\mathbb{R}^3$. In this case, one possible $H$-equivariant $\bar{h}(x)$ is the point cloud centroid: $\bar{h}(x) := \frac{1}{N} \sum_{i=1}^{N} x_i$, whose inverse would be $\bar{h}(x)^{-1} = -\bar{h}(x)$. More examples are in Appendix D.2.1 and Appendix D.3.1.

When we adopt the homeomorphic manifold assumption and use a parametric model $G_\phi : \mathcal{X} \rightarrow \mathbb{R}^m$, the invariant encoder can be defined as follows: $g_\phi(x, \tau) := G_\phi\left(\left[\bar{h}(\tau)^{-1} \cdot (x, \tau)\right]_x\right)$, where $g_\phi$ needs to take $\tau$ as an input while $G_\phi$ does not.

# 4 Experiments

In this section, we compare our EMMP framework mainly with the existing manifold primitives method, TC-VAE [37], using both synthetic and real-world robot experiments. Additionally, we compare MMP, MMP + indep, EMMP, and EMMP + indep, where the EMMP uses an invariant encoder and equivariant decoder, and '+ indep' indicates adding the independence regularization term in autoencoder training. When training TC-VAE, MMP, and MMP + indep, we always apply random data augmentation on $\tau$. Throughout, we use Gaussian Mixture Model (GMM) to fit $p(z)$.

**Evaluation metrics**: We report four evaluation metrics. First, the manifold learning accuracy is measured by the Reconstruction Error (RE). Second, to measure the degree of independence, we report estimated Mutual Information (MI) between $z$ and $\tau$, by using MINE [48]. Third, we compute the probability density learning performance by the Negative Log-Likelihood (NLL) measured in the data space. Lastly, we report the task Success Rate (SR), of which criteria is task-dependent. Details on the computations of these measures are described in Appendix D.1.

## 4.1 Goal-Reaching Task of a Planar Mobile Robot

In this section, we consider a goal-reaching task for a planar mobile robot, given a cross-shaped wall, without colliding with the wall; the robot is required to enter through one of the two closest passages (see Figure 4). As shown, it is reasonable to assume a set of feasible trajectories form a continuous manifold. Our goal is to learn the motion manifold primitive for an arbitrarily rotated wall and robot's initial position.

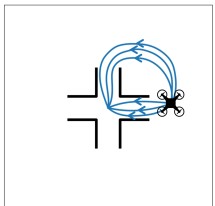

Figure 4: Demo trajectories.

The configuration space is $\mathcal{Q} = \mathbb{R}^2$, the trajectory space is $\mathcal{X} = \mathbb{R}^{2T}$. The task parameter $\tau$ consists of the initial position of the mobile robot, denoted by $(q_r^1, q_r^2)$, and the rotated angle of the wall axis $\hat{x}_w$ with respect to $\hat{x}_s$, denoted by $\omega_w$ (see Figure 5 *Left*). To make demonstration data, a human demonstrator has drawn feasible trajectories, 6 trajectories for 75 randomly given task parameters. Figure 4 illustrates example demonstration trajectories. More details on data generation and data split for training, validation, and testing are included in Appendix D.2.2.

**Symmetries:** There exist three symmetries that preserve the relative geometric relation between the wall and initial mobile robot position: (i) flipping of the robot over the wall axis $\hat{x}$, (ii) rotation of the robot around the origin by 90, 180, and 270 degrees, and (iii) rotation of the wall and mobile robot around the origin by the same amount (see Figure 5 *Right*). These symmetry transformations can

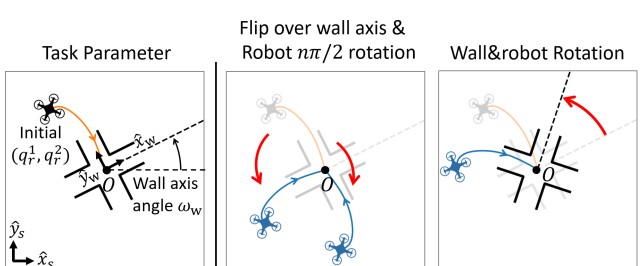

Figure 5: *Left*: The task parameter of 2D mobile robot's goal reaching task. *Right*: Symmetry transformations.

be described as group actions with the symmetry group $H := p4m \times \text{SO}(2)$, where $p4m$ is a specific type of wallpaper group and SO(2) is the group of $2 \times 2$ rotation matrices. We denote $p4m := \{(i, j) | i \in \{0, 1\}, j \in \{0, 1, 2, 3\}\}$, where $i \in \{0, 1\}$ represents flipping and

$j \in \{0, 1, 2, 3\}$ represents $n\pi/2$ rotation. Details on the group operations in $H$ and group actions on $\mathcal{T}$ and $\mathcal{X} \times \mathcal{T}$ are discussed in Appendix D.2.1.

**Construction of $\bar{h}$:** Figure 6 visualizes the equivariant map $\bar{h}(\tau) = (\bar{h}_1(\tau), \bar{h}_2(\tau))$. $\bar{h}_1(\tau) \in p4m$ is determined based on the robot's position relative to the wall state as shown in Figure 6 (*Left*). $\bar{h}_2(\tau) \in \mathrm{SO}(2)$ is determined based on the wall axis angle $\omega_w$ (see Figure 6 *Right*). The $\bar{h}(\tau)$ is equivariant to $p4m \times \mathrm{SO}(2)$; more details including the proof are in Appendix D.2.1.

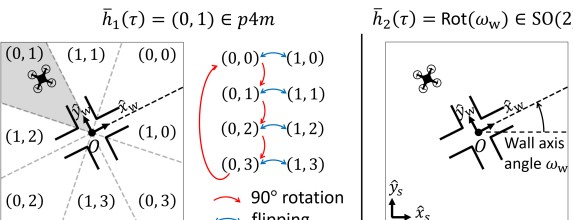

Figure 6: Illustration of the $(p4m, \mathrm{SO}(2))$-equivariant map $\bar{h}(\tau)$.

Assuming a one-dimensional latent space, we train fully connected autoencoders for MMPs and EMMPs, and TC-VAE where temporal convolutional neural networks are used as in [37]. Table 1 shows the four evaluation metrics, where the success rate is measured as follows: (i) we sample $(\tau, z)$ from the uniform distribution on $\mathcal{T}$ and learned density $p(z)$, (ii) generate trajectories via $f(z, \tau)$, and (iii) the generated trajectory is considered successful if it is consistent with the task parameter and reaches the goal without colliding to the wall. More details are in Appendix D.2.2.

Table 1: Reconstruction Error (RE), Mutual Information (MI), and Negative Log-Likelihood (NLL); the lower, the better. Success Rate (SR); the higher, the better.

| Method | RE ($\downarrow$) | MI ($\downarrow$) | NLL ($\downarrow$) | SR ($\uparrow$) |
|---|---|---|---|---|
| TC-VAE [37] | 0.257 | 0.268 | $1.50 \times 10^4$ | 53.18% |
| MMP | 0.223 | 0.487 | $1.49 \times 10^4$ | 50.08% |
| MMP + indep | 0.225 | 0.329 | $1.48 \times 10^4$ | 52.39% |
| EMMP | 0.223 | 0.082 | $1.25 \times 10^4$ | 92.40% |
| EMMP + indep | 0.229 | 0.077 | $1.24 \times 10^4$ | 86.66% |

First of all, the EMMP methods show much higher success rates than both the MMP methods and TC-VAE, which is in large part attributed to the independence between $z$ and $\tau$ and latent density fitting as shown in Table 1. As observed from the MI scores of the MMP + indep (0.329) and EMMP (0.082), using invariant and equivariant mappings is much more effective than the independence regularization term at making $z$ and $\tau$ independent, which consequently makes learning $p(z)$ easier (see Figure 7).

Second, the success rates of MMPs and TC-VAE do not show a noticeable difference. This suggests that the network architecture (temporal CNN or FCN) and autoencoder method (VAE or AE) do not have a significant impact on performance. A further comparison using EMMP in this regard is provided in Appendix D.2.3. Third, while the independence regularization term slightly improves the success rate of MMP, for EMMP, it rather shows a negative effect. Given the already low MI of EMMP, further reducing it at the expense of RE appears to have negatively impacted the success rate.

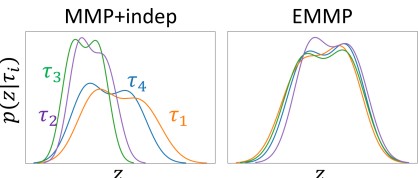

Figure 7: $p(z|\tau)$ of the MMP + indep and EMMP.

Figure 8 (*Top*) visualizes learned latent space densities $p(z)$ of TC-VAE and EMMP + indep. We select equally-spaced 6 latent points $\{z_i\}_{i=1}^{6}$ in $\mathcal{Z}$ for each model as shown in Figure 8 (blue points). The trajectories are then generated by the decoders as shown in Figure 8 (*Bottom*).

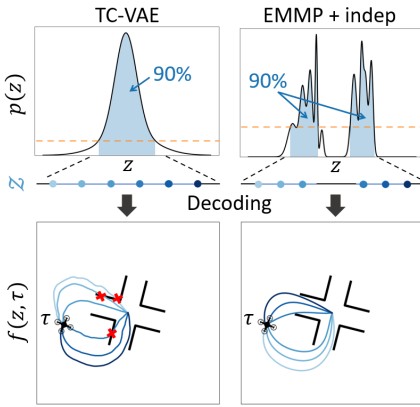

Figure 8: *Top*: $p(z)$. *Bottom*: Generated trajectories.

The EMMP + indep shows a much higher success rate than the TC-VAE; the support of $p(z)$ in our method has two connected components, each corresponding to the set of trajectories that passes through one of the two closest passages to the robot.

## 4.2  Water-Pouring Task of a Franka Panda Robot

In this section, we consider a water-pouring task for a Franka Emika Panda robot arm; the robot is assumed to hold the bottle initially upright and is required to pour 150g of water into the cup (see Figure 9 *Left*). As shown in Figure 9 *Right*, a human demonstrator provides multiple water-pouring trajectories, which are assumed to form a continuous manifold. Our goal is to learn the motion manifold primitive for an arbitrary cup position, a bottle's initial pose, and the amount of water in the bottle. In

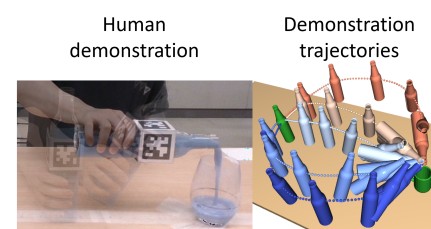

Figure 9: Demonstration data.

particular, depending on the amount of water in the bottle, the demonstration trajectories have very different characteristics (see Figure 15 *Left* in Appendix D.3.3).

The configuration space is the space of the bottle pose, $\mathcal{Q} = \mathrm{SE}(3)$, the trajectory space is $\mathcal{X} = \mathrm{SE}(3)^T$. The task parameter $\tau$ consists of the cup position on the table $(q_c^1, q_c^2) \in \mathbb{R}^2$, the bottle's initial pose $T_b \in \mathrm{SE}(3)$, and the mass of the water $m_w \in [0.2, 0.41]$ (See Figure 10 *Left*). We collect 5 demonstration trajectories for 35 different task parameters in a total of 175 trajectories (details are included in Appendix D.3.2). The demonstration trajectories are collected by recording a video of a human demonstrator performing the water-pouring task and extracting the bottle's SE(3) trajectories using AprilTag [49]. Figure 9 *Right* shows 5 trajectories demonstrated for a given single task parameter $\tau$. More details on the task parameter selection, data generation, and data split for training, validation, test are included in Appendix D.3.2.

**Symmetries**: There exist three symmetries that preserve the relative positional and geometric relationship between the cup and the bottle: (i) translation of the cup and the bottle by the same amount, (ii) rotation of the bottle around the cup, and (iii) rotation of the

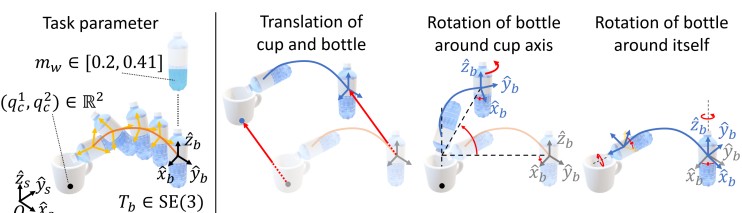

Figure 10: *Left*: The task parameter of water-pouring task. *Right*: Symmetry transformations.

bottle around itself (see Figure 10 *Right*). The symmetry transformations can be described as group actions with the symmetry group $H := \mathbb{R}^2 \times \mathrm{SO}(2) \times \mathrm{SO}(2)$. Details on the group operations in $H$ and group actions on $\mathcal{T}$ and $\mathcal{X} \times \mathcal{T}$ are discussed in Appendix D.3.2.

**Definition of $H$ and $\bar{h}$**: We define $\bar{h}(\tau) = (\bar{h}_1(\tau), \bar{h}_2(\tau), \bar{h}_3(\tau))$ as shown in Figure 11. $\bar{h}_1(\tau)$ is determined as the cup's position as shown in Figure 11 *Left*. $\bar{h}_2(\tau)$ is determined based on the position of the bottle relative to the cup (see Figure 11 *Middle*). $\bar{h}_3(\tau)$ is determined

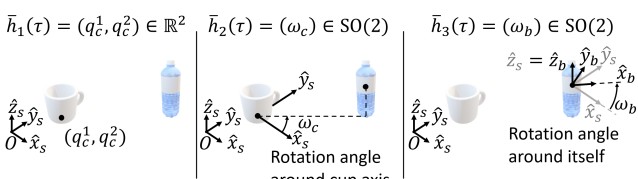

Figure 11: $(\mathbb{R}^2, \mathrm{SO}(2), \mathrm{SO}(2))$-equivariant module $\bar{h}(\tau)$.

based on the bottle's orientation relative to the base frame (see Figure 11 *Right*).

Assuming a two-dimensional latent space dimension, we train EMMP + indep with fully connected neural networks and TC-VAE. Table 2 shows the four evaluation metrics (RE, MI, NLL, SR) – where the generated robot motion is considered successful if even a little water can be poured into the cup without spilling – and one additional metric that measures the error in the amount of poured water (150g water has been poured in demonstration data), which we refer to as the Water-Pouring Error (WPE). To generate motions given a task parameter $\tau$ from $\mathcal{T}$, we sample $z$ from the learned density $p(z)$ and generate the bottle's trajectory via $f(z, \tau)$. If the generated SE(3) trajectory is out of the robot's workspace, i.e., the inverse kinematics solution does not exist, then we re-sample $z$

until we obtain a feasible trajectory. We measure the SR and WPE with 4 task parameters each with 5 samples, where we run a total of 20 generated trajectories on the real Panda robot.

As shown in Table 2, the EMMP + indep significantly outperforms TC-VAE. The large margin in RE and MI leads to lower NLL and a much higher task success rate of EMMP. Out of 20 trials of the TC-VAE, 7 trials fail to pour water into the cup and 4 trials spill water, resulting in only 9 successful pourings. On the other hand, EMMP + indep results in a 100% success rate. In addition, the WPE in EMMP + indep is also much lower than that of TC-VAE.

Table 2: Reconstruction Error (RE), Mutual Information (MI), Negative Log-Likelihood (NLL), and Water-Pouring Error (WPE); the lower, the better. Success Rate (SR); the higher, the better.

| Method | TC-VAE | EMMP + indep |
|---|---|---|
| RE ($\downarrow$) | 0.183 | 0.129 |
| MI ($\downarrow$) | 0.758 | 0.081 |
| NLL ($\downarrow$) | $1.18 \times 10^5$ | $6.40 \times 10^4$ |
| SR ($\uparrow$) | 9/20 | 20/20 |
| WPE ($\downarrow$) | $86.8 \pm 59.1$ | $23.0 \pm 11.9$ |

While the EMMP's WPE (23.0g) out of 150g seems high, we note that this error is not caused by the manifold primitive learning error, but rather is attributed to the error caused when processing AprilTags and smoothing the trajectories. Even when we replay the demonstration trajectories on the robot, the water pouring error exists and it is 19.3g on average, implying 23.0g error is not that high.

To show the strong adaptability of our framework, we perform an obstacle avoidance task using EMMP + indep. Suppose there is an obstacle, not seen during training, that blocks some water-pouring trajectories in the learned manifold primitives (e.g., Figure 12 *Left*). Since we have learned a motion manifold, not a single trajectory, even if some trajectories are blocked, we can easily find an alternative collision-free trajectory from the learned manifold primitives as shown in Figure 12 *Right*. More details on the collision detection and obstacle avoidance algorithms are in Appendix D.3.2

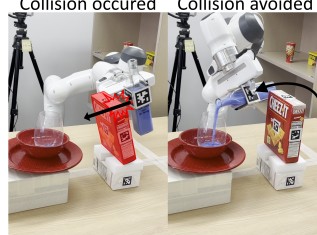

Collision occured    Collision avoided

Figure 12: Obstacle avoidance.

## 5  Limitations

One of the key assumptions in our framework is the homeomorphic manifold assumption, that is $\mathcal{M}_\tau$ for all $\tau \in \mathcal{T}$ are homeomorphic, which may not hold depending on the problem. In such cases, instead of $p(z)$, we need to fit $p(z|\tau)$, which would require more demonstration data. Second, to construct an invariant encoder and equivariant decoder, we need to define an equivariant map $\bar{h}$ for a given symmetry group $H$. Although, in our case studies, it is relatively straightforward to construct $\bar{h}$, this process may not be trivial or even impossible depending on the problem. Lastly, as our experimental results show, the independence regularization term itself is not sufficient to enforce independence between $z$ and $\tau$. Finding a better independence regularization method would be an important future research direction.

## 6  Conclusion

In this paper, we have proposed a new family of highly adaptable movement primitive models, motion manifold primitives – which is a set of trajectory manifolds $\{\mathcal{M}_\tau\}$ for all task parameters $\tau$ –, and an autoencoder-based framework for learning them. To tackle the challenges in learning $\mathcal{M}_\tau$ such as requiring many demonstration data, (i) under the homeomorphic manifold assumption, we develop the motion manifold primitives framework and introduce the independence regularization term – where we enforce independence between $z$ and $\tau$ so that it is sufficient to learn $p(z)$ instead of $p(z|\tau)$ – and (ii) we propose equivariant motion manifold primitives for an arbitrary symmetry group $H$ in the robot task and a method to parameterize it by constructing an invariant encoder and equivariant decoder. Extensive experiments have confirmed the strong adaptability of our framework and that the equivariant manifold modeling is highly effective at learning accurate $\mathcal{M}_\tau$, which leads to superior performance compared to the existing method by a significant margin.

**Acknowledgments**

B. Lee, S. Kim, and F. C. Park were supported in part by SRRC NRF grant RS-2023-00208052, IITP-MSIT grant 2021-0-02068 (SNU AI Innovation Hub), IITP-MSIT grant 2022-0-00480 (Training and Inference Methods for Goal-Oriented AI Agents), KIAT grant P0020536 (HRD Program for Industrial Innovation), ATC+ MOTIE Technology Innovation Program grant 20008547, SNU-AIIS, SNU-IAMD, SNU BK21+ Program in Mechanical Engineering, and SNU Institute for Engineering Research. Y. Lee was the beneficiary of an individual grant from CAINS supported by a KIAS Individual Grant (AP092701) via the Center for AI and Natural Sciences at Korea Institute for Advanced Study.

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

# A  Proofs of Propositions

***Proof of Group-Equivariance of (3.1)***. The equivariance of the ground truth EMMP and the invariance of $g$ proves the proposition as follows:

$$\mathcal{Z}_{h \cdot \tau} = g(\mathcal{M}_{h \cdot \tau}, h \cdot \tau) = g(\bigcup_{x \in \mathcal{M}_\tau} h \cdot (x, \tau)) = g(\bigcup_{x \in \mathcal{M}_\tau} (x, \tau)) = g(\mathcal{M}_\tau, \tau) = \mathcal{Z}_\tau. \tag{2}$$

$\square$

***Proof of Group-Equivariance of (3.2)***. The equivariance of $f$ and invariance of $g$ proves the proposition as follows:

$$\widehat{\mathcal{M}}_{h \cdot \tau} = f(\mathcal{Z}_{h \cdot \tau}, h \cdot \tau) = \bigcup_{z \in \mathcal{Z}_\tau} \left[ h \cdot (f(z, \tau), \tau) \right]_x = \bigcup_{x \in \widehat{\mathcal{M}}_\tau} \left[ h \cdot (x, \tau) \right]_x = \{ \left[ h \cdot (x, \tau) \right]_x \mid x \in \widehat{\mathcal{M}}_\tau \}. \tag{3}$$

$\square$

***Proof of Group-Equivariance of (3.3)***. Invariance of $g_\phi$ can be seen by the equivariance of $\bar{h}$, as follows:

$$g_\phi(h \cdot (x, \tau)) = G_\phi(\bar{h}(h \cdot \tau)^{-1} \cdot (h \cdot (x, \tau))) = G_\phi((h\bar{h}(\tau))^{-1}h) \cdot (x, \tau)) = G_\phi(\bar{h}(\tau))^{-1} \cdot (x, \tau)) = g_\phi(x, \tau) \tag{4}$$

Equivariance of $f_\theta$ can be seen as follows:

$$\begin{aligned} f_\theta(z, h \cdot \tau) &= \left[ \bar{h}(h \cdot \tau) \cdot (F_\theta(z, \bar{h}(h \cdot \tau)^{-1} \cdot (h \cdot \tau)), \bar{h}(h \cdot \tau)^{-1} \cdot (h \cdot \tau)) \right]_x \\ &= \left[ h \cdot \bar{h}(\tau) \cdot (F_\theta(z, \bar{h}(\tau)^{-1}(\tau)), \bar{h}(\tau)^{-1} \cdot \tau) \right]_x \\ &= \left[ h \cdot (f_\theta(z, \tau), \tau) \right]_x \end{aligned} \tag{5}$$

$\square$

# B  Related Works

In this section, we provide an overview of areas related to our work.

## B.1  Movement Primitives

In this section, we consider any form of mathematical representation used to describe motions (e.g., trajectories) that perform a given task—as specified by a task parameter variable $\tau$—as movement primitives. Dynamic movement primitives encode motion trajectories in the form of time-dependent nonlinear dynamical systems. These systems consist of mass-spring-damper systems and parametric force terms, with their task parameters defined by the initial and final configurations. [4, 5, 6, 7, 8, 9, 10, 11, 12, 13]. Stable dynamical system-based approaches use state-dependent dynamical systems that are globally asymptotically stable [14, 15, 16, 17, 18, 19, 20, 21, 22]. In these dynamical systems-based methods, the initial and goal configurations can be considered as the task parameters, and the solution trajectory that connects the two configurations is the motion described by those systems. ProMP [30] represents motions as a distribution over trajectories. By conditioning the distribution with the initial configuration, a motion trajectory is achieved. In ProMP, the initial and final configurations can be considered as task parameters. TP-GMM [26] tries to adapt to unseen task parameters by encoding trajectories as GMM seen from multiple frames. The task parameters of TP-GMM are the frames of GMMs. Given unseen frames, TP-GMM generates trajectories by calculating joint distributions between GMMs. [23, 24, 25] parameterize the demonstration trajectories using Gaussian Pross Regression, and [28, 27, 29] represents the motions using GMR (no task-parameterization exists).

However, the task parameters of most movement primitives are strictly restricted to the initial and final configurations. This limits the range of tasks that can be parameterized. In the case of water pouring, the cup's position and the amount of water can not be represented by the initial and final configurations. By adopting conditional variational autoencoder's structure, MMPs and EMMPs provide freedom of defining task parameters.

### B.2 Autoencoder-Based Manifold Learning

Autoencoders have gained prominence in recent years for identifying and generating samples from a given data distribution's underlying low-dimensional manifold structure. The main reason autoencoders are frequently adopted for manifold learning is that they learn the latent space coordinates along with the manifolds. To learn more accurate manifolds, researchers have introduced additional regularization terms [38, 39, 40, 42, 43, 41, 50, 44]. For a specific structure of conditional variational autoencoder, where the decoder gets an additional conditional parameter, the need to disentangle the conditional inputs and latent values has risen. [51] introduced a regularization term to disentangle input spaces of its decoder, by solving adding an auxiliary neural network to estimate conditional inputs from latent values, and regularizing the autoencoder by making it harder for the auxiliary network to estimate. However, unlike the independence regularization term that we introduced, the regularization term does not necessarily guarantee independence between the two spaces.

#### B.2.1 Autoencoder-Based Motion Manifold Primitives

In this section, we introduce an existing motion manifold primitive framework called TC-VAE [37]. TC-VAE aims to parameterize the motion manifold given a task parameter based on autoencoder frameworks. As TC-VAE adopts the structure of [51], the decoder of it takes additional task parameter inputs other than the latent value inputs. TC-VAE also adopts the regularization term of [51] for disentangling the task parameters and the latent values, which still shares the shortcoming of not guaranteeing independence between latent space and the conditional input space.

### B.3 Equivariant Models in Robotics

Invariance and equivariance properties have played a role in deep learning models as an inductive bias to generalize well and be trained data efficiently [52]. Translation equivariance in convolutional neural networks (CNNs) has been effective for image recognition tasks [53]. Group equivariant CNNs have expanded the equivariance in CNNs to more complex equivariance, e.g. SO(3)-equivariance achieved by Spherical CNNs [54, 55]. In robot manipulation tasks, [56] proposed an SE(3)-equivariant object representation, and [45] introduced a SE(2)-equivariant dynamics model learning for pushing manipulation. Most of the existing equivariant models in robotics are restricted to certain types of groups, whereas our work can be applied to tasks with arbitrary group symmetries.

## C  Geometric Preliminaries

This section provides preliminary knowledge of geometric tools used in the paper.

### C.1  Manifold Hypothesis

Real-world observations often require a large number of variables to represent numerically. For example, an SE(3)-trajectory of length 500 lives in a high-dimensional data space $\mathbb{R}^{8000}$. Dealing with such high-dimensional data is very challenging as the amount of data needed grows exponentially with the dimensionality, known as the curse of dimensionality.

The manifold hypothesis states that high-dimensional data (e.g. trajectory) approximately lie on some lower-dimensional manifold embedded in high-dimensional space, which suggests that the high-dimensional data can be in fact described by a relatively small number of variables. For example, to describe points on a two-dimensional sphere – which are represented as unit vectors in $\mathbb{R}^3$ –, we only need two variables, e.g., the spherical $(\theta, \phi)$-coordinates.

Of particular relevance to our paper, consider a set of length $n$ trajectories in $\mathbb{R}^2$, which start from the robot and end at the star, as shown in Figure 13 (i.e., trajectories A, B, C, D, E). We note that these five trajectories are elements in the high-dimensional trajectory data space $\mathbb{R}^{2 \times n}$. However, it is clear that they do not fill up the entire space. Rather it is suspected that they form a lower-dimensional space. Each trajectory may approximately be represented with one variable that indicates how much

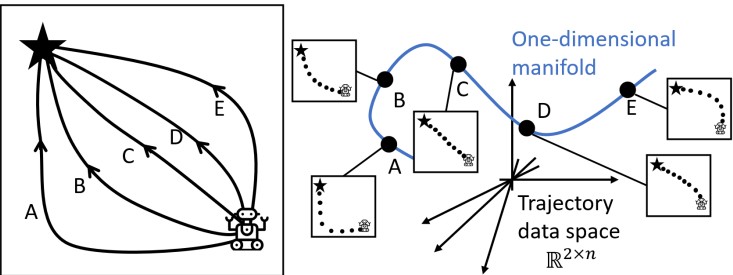

Figure 13: Trajectories in high-dimensional data space lie on a one-dimensional manifold.

it bends down or up compared to the straight line between the robot and the star, meaning that these five trajectories approximately lie on a one-dimensional manifold.

## C.2    Homeomorphism

A homeomorphism is a continuous, bijective function that has a continuous inverse function between two topological spaces (for this paper, two manifolds). Two manifolds are said to be homeomorphic if there exists a homeomorphism between the two manifolds. Intuitively, two manifolds are home-omorphic if one can be smoothly deformed into another. For example, a sphere can be smoothly deformed into an ellipsoid, hence a sphere and an ellipsoid are homeomorphic. However, there is no way to smoothly deform a sphere into a torus, which indicates that a sphere and a torus are not homeomorphic.

Suppose that there exist two $m$-dimensional manifolds $\mathcal{M}_1$ and $\mathcal{M}_2$. Let the latent space of $\mathcal{M}_1$ be $\mathcal{Z}_1 \subseteq \mathbb{R}^m$ and the coordinate space of $\mathcal{M}_2$ be $\mathcal{Z}_2 \subseteq \mathbb{R}^m$. Let $f_1 : \mathcal{Z}_1 \to \mathcal{M}_1$ and $f_2 : \mathcal{Z}_2 \to \mathcal{M}_2$ be invertible maps satisfying $f_1(\mathcal{Z}_1) = \mathcal{M}_1$ and $f_2(\mathcal{Z}_2) = \mathcal{M}_2$. If $\mathcal{M}_1$ and $\mathcal{M}_2$ are homeomorphic, i.e. there exists $g : \mathcal{M}_1 \to \mathcal{M}_2$ such that $g(\mathcal{M}_1) = \mathcal{M}_2$, $\mathcal{Z}_2 = (f_2^{-1} \circ g \circ f_1)(\mathcal{Z}_1)$ is satisfied. $\mathcal{M}_1$ and $\mathcal{M}_2$ can have shared latent space by replacing $f_2$ as $f_2'(\cdot) := (g \circ f_1)(\cdot)$.

## C.3    Group and Group Action

A group $H$ is a non-empty set, combined with a group operation $* : H \times H \to H$, where we simply denote it by $h_1 * h_2 = h_1 h_2$. A group $H$ must satisfy four conditions:

- The group $H$ contains an identity $e$.
- The group $H$ contains inverses, i.e., $h^{-1}h = hh^{-1} = e$ for all $h \in H$.
- The group operation is associative, i.e., $h_1(h_2 h_3) = (h_1 h_2)h_3$ for all $h_1, h_2, h_3 \in H$.
- The group $H$ is closed under operation, i.e., $h_1 h_2 \in H$ for all $h_1, h_2 \in H$.

A group action $\cdot : H \times \mathcal{X} \to \mathcal{X}$ is a function defined on the product space of a group and a set satisfying two conditions:

$$e \cdot x = x,$$
$$h_1 \cdot (h_2 \cdot x) = (h_1 h_2) \cdot x \text{ for all } h_1, h_2 \in H,$$

where $e \in H$ is the identity.

## C.4    Equivariance and Invariance

Given a group $H$, a function $f : X \to Y$ and group actions $\cdot$ defined on $X$ and $Y$, the function $f$ is said to be $H$-equivariant if it satisfies

$$f(h \cdot x) = h \cdot f(x),$$

for all $x \in X$ and $h \in H$. $f$ is said to be $H$-invariant if it satisfies

$$f(h \cdot x) = f(x),$$

for all $x \in X$ and $h \in H$. Invariance is a special case of equivariance where the group actions defined on $Y$ is the identity function, i.e. $h \cdot y = y$ for all $h \in H$ and $y \in Y$.

# D  Experimental and Implementation Details

Throughout the experiments, we have used RTX 2080 Ti, RTX 3080 Ti, RTX 3090 for training the models, and each experiment takes a few hours to 10 hours depending on the model.

## D.1  Evaluation Metrics

**Recontruction Error:** We measure reconstruction error in the test dataset using following equation:

$$\text{Reconstrution loss} = \sqrt{\frac{1}{N} \sum_{i=1}^{N} \frac{1}{M_i} \sum_{j=1}^{M_i} \frac{1}{T} d_{\mathcal{X}}^2(f_\theta(g_\phi(x_{ij}, \tau_i), \tau_i), x_{ij})}. \tag{6}$$

**Latent-Task Dataset:** To calculate mutual information and negative log-likelihood, we define a dataset of $(z, \tau)$ paired dataset. To build the dataset large enough, we first randomly augment $(x, \tau)$ pairs in the training dataset 100 times. Then, the every $z$ is the encoded values from $(x, \tau)$; $z = g_\phi(x, \tau)$.

**Mutual Information:** Mutual information between $z$ and $\tau$ is measured using Mutual Information Neural Estimator (MINE) [48]. MINE estimates by, which estimates mutual information by maximizing its lower bound, using the Donsker-Varadhan representation:

$$D_{KL}(\mathcal{Z}||\mathcal{T}) \geq \sup_{F \in \mathcal{F}} \mathbb{E}_{\mathcal{Z}}[F] - \log\left(\mathbb{E}_{\mathcal{T}} e^F\right), \tag{7}$$

where $\mathcal{F}$ is any class of functions $F : \Omega \to \mathbb{R}$. In our case, $\Omega = \mathcal{Z} \times \mathcal{T}$. By replacing $\mathcal{F}$ by parametric family $\mathcal{F}_\Theta$, the mutual information is estimated as follows:

$$I_\Theta(\mathcal{Z}, \mathcal{T}) = \sup_{\theta \in \Theta} \mathbb{E}_{p(z,\tau)} F_\theta - \log\left(\mathbb{E}_{p(z)p(\tau)} e^{F_\theta}\right). \tag{8}$$

We train MINE using the latent-task dataset for 1,500 iterations with a batch size of 5,000 equally for all models.

**Negative Log-Likelihood:** Given $(z, \tau)$ from the latent-task dataset, we calculate negative log-likelihood $-\log(p_{\mathcal{M}_\tau}(g_\phi(z)))$ in trajectory space $\mathcal{X}$, using the following equation:

$$p_{\mathcal{M}_\tau}(g_\phi(z, \tau)) = p_{\mathcal{Z}_\tau}(z)|\det[J_{g_\phi}^T J_{g_\phi}]|^{-\frac{1}{2}}, \tag{9}$$

where $J_{g_\phi}$ denotes $\frac{\partial g_\phi}{\partial z}(z, \tau)$.

## D.2  Planar Mobile Robot Experiment

### D.2.1  Formulas and Proofs

**Task Parameter Space:** Recall that the configuration space is $\mathcal{Q} = \mathbb{R}^2$, the trajectory space is $\mathcal{X} = \mathbb{R}^{2T}$. The task parameter $\tau$ consists of the initial position of the mobile robot, denoted by $(q_r^1, q_r^2)$, and the rotated angle of the wall axis $\hat{x}_w$ with respect to $\hat{x}_s$, denoted by $\omega_w$.

To uniformly sample from $\mathcal{T}$ we make $\mathcal{T}$ compact by restricting the initial position of the motile robot to be on a disk whose center is the origin, the inner radius is 5, and the outer radius is 10. Also, we have limited the wall rotation axis to $-\frac{\pi}{4} \leq \omega_w < \frac{\pi}{4}$ to make the problem easier for non-equivariant methods e.g. TC-VAE. since the wall's geometries are identical every 90 degrees, the wall still can span all possible geometrical configurations.

**Trajectory Space and Distance Measure:** We set the length of trajectories $T = 201$, which makes the trajectory space $\mathcal{X} = \mathbb{R}^{402}$. The distance measure is defined as: $d_{\mathcal{X}}(x_1, x_2) := ||x_1, x_2||_2$.

**Group Operations:** Recall that the symmetry group $H$ is $H := p4m \times SO(2)$, where $p4m$ is a specific type of wallpaper group and $SO(2)$ is the group of $2 \times 2$ rotation matrices. We denote $p4m := \{(i,j)|i \in \{0,1\}, j \in \{0,1,2,3\}\}$, where $i \in \{0,1\}$ represents flipping and $j \in \{0,1,2,3\}$ represents $n\pi/2$ rotation. Throughout this section, we represent an $SO(2)$ element $R = \begin{bmatrix} \cos\alpha & -\sin\alpha \\ \sin\alpha & \cos\alpha \end{bmatrix} \in SO(2)$ simply as $\alpha$.

Given two group elements $((a,b),\alpha), ((c,d),\beta) \in p4m \times SO(2)$, the group operation is defined as:

$$((a,b),\alpha)((c,d),\beta) = ((\text{mod}(a+c,2), \text{mod}(b+(-1)^a d, 4)), \alpha + \beta), \tag{10}$$

where $\text{mod}(x,y)$ denotes the remainder of $\frac{x}{y}$.

**Group Actions:** The procedure of the group actions of $H = p4m \times SO(2)$ can be explained as follows: (i) flip the robot over the wall axis (ii) rotate the robot around the origin $\frac{n\pi}{2}$, and finally, (iii) rotated the robot and the wall. Given a task parameter $\tau = (q_r^1, q_r^2, \omega_w)$ and a group element $h = ((a,b),\alpha)$, the group action $h \cdot \tau$ is then defined as:

$$h \cdot \tau = (\text{Rot}(\alpha + \omega_w + \frac{b\pi}{2}) * \text{flip}(a) * \text{Rot}(-\omega_w) * (q_r^1, q_r^2), \alpha + \omega_w) \tag{11}$$

where $*$ denotes matrix multiplication, $\text{flip}(a) := \begin{bmatrix} 1 & 0 \\ 0 & -1 \end{bmatrix}^a$, and $\text{Rot}(\alpha) := \begin{bmatrix} \cos\alpha & -\sin\alpha \\ \sin\alpha & \cos\alpha \end{bmatrix}$.

Given a trajectory $x = \{(q_i^1, q_i^2)\}_{i=1}^T$, a task parameter $\tau = (q_r^1, q_r^2, \omega_w)$ and a group element $h = ((a,b),\alpha)$, the group action $h \cdot (x, \tau)$ is defined as :

$$h \cdot (x, \tau) = (\{\text{Rot}(\alpha + \omega_w + \frac{b\pi}{2}) * (q_i^1, q_i^2)\}_{i=1}^T, h \cdot \tau). \tag{12}$$

**Group-Equivariant Map $\bar{h}$:** Given a task parameter $\tau$, $\bar{h}(\tau)$ can be divided into two elements, $\bar{h}(\tau) = (\bar{h}_1(\tau), \bar{h}_2(\tau))$, where $\bar{h}_1(\tau) = (\bar{h}_1^1(\tau), \bar{h}_1^2(\tau)) \in p4m$ and $\bar{h}_2(\tau) \in SO(2)$. Figure 14 illustrates $\bar{h}(\tau)$, and its equivariance for $\bar{h}_1$ using two group actions $h_{\text{flip}} = (1,0,0)$ and $h_{\text{rot90}} = (0,1,0)$. It can be seen by the commutative diagram that $\bar{h}(h_{\text{flip}} \cdot \tau) = h_{\text{flip}}\bar{h}(\tau)$ and $\bar{h}(h_{\text{rot90}} \cdot \tau) = h_{\text{rot90}}\bar{h}(\tau)$. The rest cases of flipping and rotating 90, 180, 270 degrees can be shown in the same way. As shown, $\bar{h}_2(\tau)$ is defined as $\bar{h}_2(\tau) = \omega_w$. The equivariance of $\bar{h}_2$ can be shown by $h_{\text{rot}} = (0,0,\alpha)$ and $\tau = (q_r^1, q_r^2, \omega_w)$:

$$\bar{h}(h_{\text{rot}} \cdot \tau) = (q_r^1, q_r^2, \omega_w + \alpha) = (0,0,\alpha)(q_r^1, q_r^2, \omega_w) = h_c \bar{h}(\tau). \tag{13}$$

Equivariance for the case where $h = (a,b,\alpha)$ is then simply shown by dividing it into $h = (0,0,\alpha)(a,b,0)$:

$$\begin{aligned}
\bar{h}((a,b,\alpha) \cdot \tau) &= \bar{h}(((0,0,\alpha)(a,b,0)) \cdot \tau) \\
&= \bar{h}((0,0,\alpha) \cdot (a,b,0) \cdot \tau) \\
&= (0,0,\alpha)\bar{h}((a,b,0) \cdot \tau) \\
&= (a,b,0)(0,0,\alpha)\bar{h}(\tau) \\
&= (a,b,\alpha)\bar{h}(\tau).
\end{aligned} \tag{14}$$

Below equation is the formal definition of $\bar{h}$ given a $\tau = (q_r^1, q_r^2, \omega_w)$:

$$\bar{h}_1^1(\tau) = \begin{cases} 0, & \text{if } \frac{k\pi}{2} \leq \text{atan2}(q_r^2, q_r^2) - \omega_w < \frac{k\pi}{2} + \frac{\pi}{4} \\ 1, & \text{otherwise} \end{cases},$$

$$\bar{h}_1^2(\tau) = \begin{cases} 0, & \text{if } -\frac{\pi}{4} \leq \text{atan2}(q_r^2, q_r^2) - \omega_w < \frac{\pi}{4} \\ 1, & \text{if } \frac{\pi}{4} \leq \text{atan2}(q_r^2, q_r^2) - \omega_w < \frac{3\pi}{4} \\ 2, & \text{if } \frac{3\pi}{4} \leq \text{atan2}(q_r^2, q_r^2) - \omega_w < \pi \\ & \text{or } -\pi \leq \text{atan2}(q_r^2, q_r^2) - \omega_w < -\frac{3\pi}{4} \\ 3, & \text{if } -\frac{3\pi}{4} \leq \text{atan2}(q_r^2, q_r^2) - \omega_w < -\frac{\pi}{4} \end{cases}, \tag{15}$$

$$\bar{h}_2(\tau) = \omega_w,$$

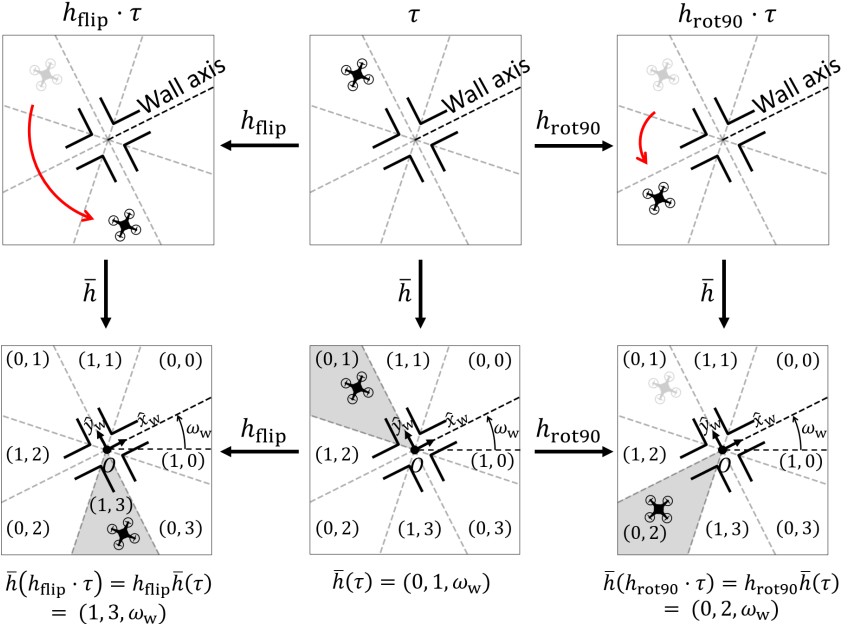

Figure 14: Illustration of $\bar{h}$ and its equivariance. $h_{\text{flip}} := (1,0,0)$ is the flipping motion of the mobile robot over the wall axis, and $h_{\text{rot90}} := (0,1,0)$ is the rotation of the mobile robot 90 degrees around the origin. It can be seen that $h_{\text{flip}}\bar{h}(\tau) = (1,0,0)(0,1,\omega_w) = (1, \text{mod}(0-1,4), \omega_w) = (1,3,\omega_w) = \bar{h}(h_{\text{flip}}\cdot\tau)$ and $h_{\text{rot90}}\bar{h}(\tau) = (0,1,0)(0,1,\omega_w) = (0,\text{mod}(1+1,4,\omega_w) = (0,2,\omega_w) = \bar{h}(h_{\text{rot90}}\cdot\tau)$.

where $k \in \{-2,-1,0,1\}$ and $(\text{atan2}(q_r^2, q_r^2) - \omega_w)$ is assumed to be satisfying $-\pi \leq \text{atan2}(q_r^2, q_r^2) - \omega_w < \pi$.

### D.2.2 Experimental Details

**Datasets:** For dataset generation, we first uniformly sample from the smallest space that can span $\mathcal{T}$ by symmetry transformations, in which the robot's initial position $q_r$ satisfies $5 \leq ||q_r|| < 10$ and $0 \leq \text{atan2}(q_r^2, q_1^1) < \pi/4$, and the wall axis angle is 0. We collect trajectory data by generating B-splines given via points labeled by humans. The B-splines are then reparameterized so that the time length of the splines becomes 5 seconds, where the splines accelerate for the first second and decelerate for the last second. We finally sample 201 points from the splines.

For the training dataset, we have gathered 6 trajectories for 75 randomly given task parameters, a total of 300 trajectories for training. For the validation dataset, we have gathered a trajectory for 80 randomly given task parameters and randomly augmented them using symmetry transformations 100 times. For the test dataset, we have gathered a trajectory for 40 randomly given task parameters and randomly augmented them using symmetry transformations 1000 times. The number of validation and test datasets are then 8,000 and 40,000 repectively.

**Network Architectures and Training Details:** A task parameter $\tau$ is represented as $(q_r^1, q_r^2, \omega_w)$, where $(q_r^1, q_r^2)$ is the mobile robot's initial position and $\omega_w$ is the wall's axis angle. In practical implementation, we use $(q_r^1, q_r^2, \cos\omega_w, \sin\omega_w) \in \mathbb{R}^4$ as an input parameter vector. Since $T = 201$, the output space is $\mathbb{R}^{402}$.

We use two-layer fully connected neural networks of 512 nodes for MMPs and EMMPs with elu as their activation function. TC-VAE's encoder includes a fully connected network and a temporal convolutional network, and the decoder includes two fully connected networks for $z$ and $\tau$, a temporal convolutional network, and a fully connected network. All four fully connected networks used in TC-VAE are of two layers with size 434. The output sizes of fully connected networks for $z$ and $\tau$ in the decoder are 36 and 72 respectively. The two temporal convolutional layers in TC-VAE are

both with channel sizes $(18, 36, 72)$ and kernel size 3. More details on the structure of TC-VAE are in [37]. All models in the experiments have similar number of parameters ($\approx 9.4 \times 10^5$).

**Success Criterion:** We consider a trajectory successful if it is consistent with the task parameter and reaches the goal without colliding with the wall. More specifically, we check (i) collision avoidance, (ii) the robot's initial position, and the robot's final position. We consider the trajectory satisfies (ii) and (iii) if the initial configuration and the final configuration are within a radius of $0.3$ at the initial position specified in the task parameter and origin, respectively. The number of sample $(z, \tau)$ we use for success rate calculation is 50,000.

### D.2.3  Additional results

**Architecture Comparison:** We compare MMPs and EMMPs of fully connected autoencoders (denoted as AE), fully connected variational autoencoders (denoted as VAE), and variational autoencoders of the same structure with TC-VAE (denoted as TC-VAE). Table 3 shows the four evaluation metrics. Overall, as shown in the success rate scores, regardless of network architecture and autoencoder method, EMMPs without regularization perform the best, and MMPs without regularization perform the worst. Although EMMP (TC-VAE) excels in most measures (MI and NLL) its success rate (91.2%) is still lower than EMMP (AE)'s (92.40%) and EMMP (VAE)'s (95.72%), which is caused by the tendency of (TC-VAE) that it violates the initial and final condition in about $6\%$ of trials, whereas EMMP (AE) only violates them and EMMP (VAE) almost never violate them (0% $\sim 0.01\%$).

**Equivariance Comparison:** Here, we qualitatively compare the equivariance performance of random data augmentation and equivariant learning method by comparing MMP (AE) and EMMP (AE). Figure 15 shows trajectories generated by $\tau$ and $h \cdot \tau$, with same $z$. If the decoder $f$ is equivariant, $f(z, h \cdot \tau)$ (blue lines in the figure) and $[h \cdot (f(z, \tau), \tau)]_x$ (grey lines in the figure) must overlap. However, as shown in the Figure 15 *Left*, trajectories of the MMP do not , whereas trajectories of the EMMP perfectly overlap as shwon in Figure 15 *Right*.

## D.3  Water-Pouring Experiment

### D.3.1  Formulas and Proofs

**Task Parameter Space:** The input space $Q = \mathbb{R}^2 \times \mathrm{SE}(3) \times [0.2, 0.41]$. A task parameter $\tau$ can be represented as $((q_c^1, q_c^2), (q_b^1, q_b^2, q_b^3, R_b)), m_w)$, where $(q_c^1, q_c^2)$ is the cup's position, $(q_b^1, q_b^2, q_b^3, R_b))$ is the bottle's initial position and orientation, and $m_w$ is the weight of water in the bottle. To construct compact $\mathcal{T}$, we limit $(q_c^1, q_c^2)$ to be inside a square at the origin with edge length of $0.5$, i.e., $-0.25 \leq q_c^1, q_c^2 \leq 0.25$, and limit the distance between $(q_c^1, q_c^2)$ and $(q_b^1, q_b^2)$ to satisfy $0.3 \leq ||q_b^1 - q_c^1, q_b^2 - q_b^2||_2 \leq 0.78$. Since the bottle is on the table upright, $q_b^3$ is a constant.

**Trajectory Space and Distance Measure:** We set the length of trajectories $T$ to be 480, which makes trajectory space $\mathcal{X} = \mathrm{SE}(3)^{480}$. Given $x_1 = \{x_1^i\}_{i=1}^{480}$ and $x_2 = \{x_2^i\}_{i=1}^{480}$, where each $x_j^i$ can be represented by $(R_{ij} \in \mathrm{SO}(3), p_{ij} \in \mathbb{R}^3)$, the distance measure on $\mathcal{X}$ is defined as:

$$d_{\mathcal{X}}(x_1, x_2) := \sqrt{\sum_i \left( ||R_{i1}^{-1} * R_{i2} - I||_F^2 + \gamma ||p_{i1} - p_{i2}||_2^2 \right)}, \tag{16}$$

Table 3: Reconstruction Error (RE), Mutual Information (MI), and Negative Log-Likelihood (NLL); the lower, the better. Success Rate (SR); the higher, the better.

| Method | RE ($\downarrow$) | MI ($\downarrow$) | NLL ($\downarrow$) | SR ($\uparrow$) |
|---|---|---|---|---|
| MMP (AE) | 0.223 | 0.487 | $1.49 \times 10^4$ | 50.08% |
| MMP (VAE) | 0.233 | 0.687 | $1.57 \times 10^4$ | 42.98% |
| MMP (AE) + indep | 0.225 | 0.329 | $1.48 \times 10^4$ | 52.39% |
| MMP (VAE) + indep | 0.229 | 0.652 | $1.56 \times 10^4$ | 44.28% |
| EMMP (AE) | 0.223 | 0.082 | $1.25 \times 10^4$ | 92.40% |
| EMMP (AE) + indep | 0.229 | 0.077 | $1.24 \times 10^4$ | 86.66% |
| EMMP (VAE) | 0.231 | 0.066 | $1.23 \times 10^4$ | 95.72% |
| EMMP (VAE) + indep | 0.225 | 0.167 | $1.28 \times 10^4$ | 82.02% |
| EMMP (TC-VAE) | 0.227 | 0.065 | $1.00 \times 10^4$ | 91.20% |
| EMMP (TC-VAE) + indep | 0.247 | 0.071 | $1.15 \times 10^4$ | 88.22% |

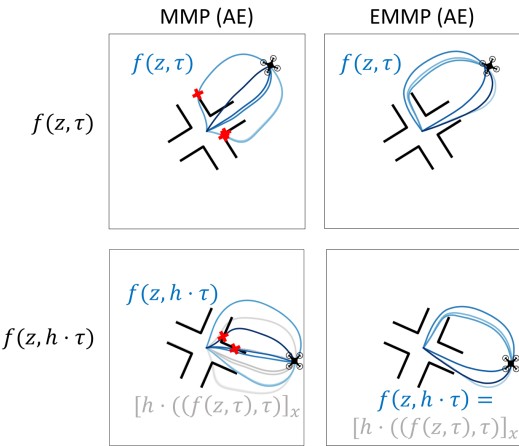

Figure 15: Equivariance comparison between MMP (AE) and EMMP (AE). If the decoder $f$ is equivariant, $f(z, h \cdot \tau)$ (blue lines in the figure) and $[h \cdot (f(z, \tau), \tau)]_x$ (grey lines in the figure) must overlap. It can be seen that the decoder of MMP is not equivariant, whereas the EMMP' decoder is equivariant.

where $\gamma = 5$ is a constant.

**Group Operations:** The symmetry group $H = \mathbb{R}^2 \times \mathrm{SO}(2) \times \mathrm{SO}(2)$, each is for translation of the cup and the bottle, rotation of the bottle around the cup, and rotation of the bottle around itself. Given two group elements $(a, b, R_{c1}, R_{b1}), (c, d, R_{c2}, R_{b2}) \in \mathbb{R}^2 \times \mathrm{SO}(2) \times \mathrm{SO}(2)$, the group operation is defined as follows:

$$(a, b, R_\alpha, R_\beta)(c, d, R_\gamma, R_\delta) = (a + c, b + d, R_\alpha * R_\gamma, R_\beta * R_\delta). \tag{17}$$

**Group Actions:** The procedure of group actions of $h \in H$ can be explained as follows: (i) translate the cup and the bottle, (ii) rotate the bottle around the cup, and (iii) rotate the bottle around itself. Given a task parameter $\tau = ((q_c^1, q_c^2), (q_b^1, q_b^2, q_b^3, R_b)), m_w)$, and a group element $h = (a, b, R_\alpha, R_\beta)$, the group action $h \cdot \tau$ is defined as:

$$h \cdot \tau = \left( (q_c^1 + a, q_c^2 + b), ((q_c^1 + a, q_c^2 + b, 0) + (R_\alpha * (q_b^1 - q_c^1, q_b^2 - q_c^2, q_b^3)), R_\alpha * R_b * R_\beta), m_w \right). \tag{18}$$

**Group-Equivariant Map $\bar{h}$:** Given a task parameter $\tau = ((q_c^1, q_c^2), (q_b^1, q_b^2, q_b^3, R_b)), m_w)$, $\bar{h}(\tau) = (\bar{h}_1(\tau), \bar{h}_2(\tau), \bar{h}_2(\tau))$ is defined as follows:

$$\bar{h}_1(\tau) = (q_c^1, q_c^2) \in \mathbb{R}^2, \tag{19}$$

$$\bar{h}_2(\tau) = \mathrm{Rot}(\hat{z}, \theta_1), \tag{20}$$

$$\bar{h}_3(\tau) = \mathrm{Rot}(\hat{z}, (\theta_2 - \theta_1)), \tag{21}$$

where $\theta_1 := \mathrm{atan2}(q_b^2 - q_c^2, q_b^1 - q_c^1)$, $\theta_2 := \mathrm{atan2}(\hat{x}_b^2, \hat{x}_b^1)$, and $\hat{x}_b$ denotes the first column of $R_b$.

Given an arbitrary $h = (a, b, R_\alpha, R_\beta)$, where $R_\alpha = \mathrm{Rot}(\hat{z}, \alpha)$ and $R_\beta = \mathrm{Rot}(\hat{z}, \beta)$, the equivariance of $\bar{h}$ is shown by the following equation:

$$
\begin{aligned}
\bar{h}(h \cdot \tau) &= \bar{h}((q_c^1 + a, q_c^2 + b), ((q_c^1 + a, q_c^2 + b, 0) + (R_\alpha * (q_b^1 - q_c^1, q_b^2 - q_c^2, q_b^3)), R_\alpha * R_b * R_\beta), m_w) \\
&= ((q_c^1 + a, q_c^2 + b), \mathrm{Rot}(\hat{z}, \alpha + \theta_1), \mathrm{Rot}(\theta_2 - \theta_1 + \beta) \\
&= ((q_c^1 + a, q_c^2 + b), R_\alpha * \mathrm{Rot}(\hat{z}, \theta_1), \mathrm{Rot}(\hat{z}, \theta_2 - \theta_1) * R_\beta) \\
&= ((q_c^1 + a, q_c^2 + b), R_\alpha * \mathrm{Rot}(\hat{z}, \theta_1), R_\beta * \mathrm{Rot}(\hat{z}, \theta_2 - \theta_1)) \\
&= (a, b, R_\alpha, R_\beta)((q_c^1, q_c^2), \mathrm{Rot}(\hat{z}, \theta_1), \mathrm{Rot}(\hat{z}, (\theta_2 - \theta_1))) \\
&= h\bar{h}(\tau). \tag{22}
\end{aligned}
$$

#### D.3.2 Experimental Details

**Datasets:** The water-pouring demonstration trajectories are collected by recording videos of water-pouring motions of a human demonstrator for 8 seconds (intended for 3.5 seconds of reaching motion and 4.5 seconds of pouring motion) at 60fps, with three AprilTags, resulting in 480 frames [49].

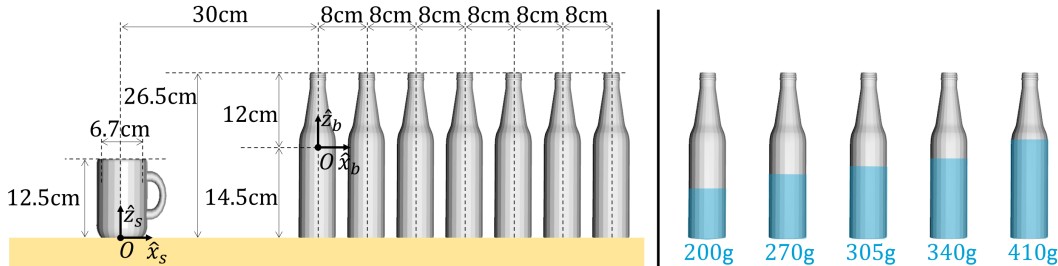

Figure 16: Task parameters for demonstration. The cup is always at the origin, and the bottle is at $(0, r \in 0.3, 0.38, 0.54, 0.62, 0.78, 0.145)$. The mass of water in the bottle is $m_w \in 0.2, 0.27, 0.305, 0.34, 0.41$.

Then we extract the SE(3) trajectories of the bottle, whose length $T = 480$. We perform trajectory smoothing and transform task parameters and the trajectories using group actions of $H$ for the cup's position to be the origin, and the bottle to be initially in the $\hat{x}_s$-direction from the bottle, and $\hat{x}_b$ to be aligned with $\hat{x}_s$. The resulting task parameters are in the form of $((0,0), (r, 0, \mathsf{h}, R, 0)), m_w)$.

Assuming the bottle to be initially on the table upright, $\mathsf{h} = 0.145$ is constant, and $R = I$. We gather 5 trajectories for 7 different $r$ and 5 different $m_w$ in a total of 175 trajectories. As shown in Figure 16 *Left*, we choose $r$ at every 8cm, from 30cm to 78cm, i.e., $r \in \{0.3, 0.38, 0.46, 0.54, 0.62, 0.7, 0.78\}$, and as shown in Figure 16 *Right* we choose $m_w \in \{0.2, 0.27, 0.305, 0.34, 0.41\}$. The dimensions of the cup and the bottle, and the position of the bottle frame are as illustrated in Figure 16 *right*. The five demonstrations of each task parameter are intended to pour water gradually from the left side of the cup and the right side of the cup.

We use 125 trajectories of $r \in 0.3, 0.38, 0.54, 0.62, 0.78$ as the training dataset, and we randomly split the other 50 trajectories into half for validation and test dataset. We randomly augment the dataset 100 and 1,000 times for validation and test dataset respectively, resulting in 2,500 validation trajectories and 25,000 test trajectories.

**Network Architectures and Training Details:** The space of the task parameters is $\mathbb{R}^2 \times \mathrm{SE}(3) \times [0.2, 0.41]$, where a task parameter can be represented in the form of $((q_c^1, q_c^2), (q_b^1, q_b^2, q_b^3, R_b)), m_w)$. Assuming that the bottle is initially on the table upright, since $q_b^3$ is a constant variable and $R_b$ can be represented as $\mathrm{Rot}\hat{z}, \theta_b$, in practical implementation, we use $(q_c^1, q_c^2, q_b^1, q_b^2, m_w, \cos \theta_b, \sin \theta_b) \in \mathbb{R}^7$ for input of the decoder.

The output of the model is an element in $\mathrm{SE}(3)^{480}$ which is not a vector space. A naive parameterization or SE(3) element (e.g. as a 12-dimensional vector) does not enforce the model outputs to satisfy SE(3) constraints. To constraint the model output space to be $\mathrm{SE}(3)^{480}$, we first set all model output sizes to be $480 \times 6 = 2880$, and add an additional layer Vec2SE3 at the end of every decoder. Given a vector $v = (v^1, \ldots, v^6) \in \mathbb{R}^6$, Vec2SE3 is defined as:

$$\mathrm{Vec2SE3} : v \mapsto \begin{bmatrix} \exp\left(\begin{bmatrix} 0 & -v_3 & v_2 \\ v_3 & 0 & -v_1 \\ -v_2 & v_1 & 0 \end{bmatrix}\right) & \begin{matrix} v^4 \\ v^5 \\ v^6 \end{matrix} \\ 0 & 1 \end{bmatrix} \in \mathrm{SE}(3).$$

We finally vectorize the first three rows of the SE(3) matrix, since the last row is constant at $(0, 0, 0, 1)$.

We use two-layer fully connected neural networks of 168 nodes for the EMMP with elu as its activation function. TC-VAE's encoder includes a fully connected network and a temporal convolutional network, and the decoder includes two fully connected networks for $z$ and $\tau$, a temporal convolutional network, and a fully connected network. All four fully connected networks used in TC-VAE are of two layers with size 512. The output sizes of fully connected networks for $z$ and $\tau$ in the decoder are 40 and 80 respectively. The two temporal convolutional layers in TC-VAE are both with channel sizes $(36, 72, 144)$ and kernel size 3. More details on the structure of TC-VAE are in [37].

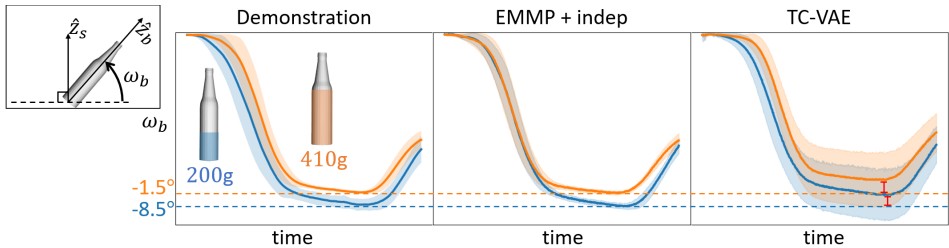

Figure 17: Graphes of bottle angle vs. time. The pouring angle $\omega_b$ is the angle between the bottle axis $\hat{z}_b$ and the $xy$-plane. The orange lines are pouring angles for the $m_w = 0.41$ case, and the blue lines are pouring angles for the $m_w = 0.20$ case. It can be observed that the pouring angle decreases as the mass of water increases.

All models in the experiments have a similar number of parameters, where EMMP contains $(1.51 \times 10^6)$ parameters and TC-VAE contains $(1.56 \times 10^6)$ parameters.

**Task Parameters for Success Rate Measure:** We sample five feasible trajectories for four task parameters. Throughout the four task parameters, the cup's position $(q_c^1, q_c^2) = (-0.2, 0)$, the bottle is initially in the $y$-direction from the cup, i.e., $q_b^1 = -0.2$, the bottle is initially aligned with the base frame, i.e., $R_b = I$. The rest parts, $(q_b^2, m_w)$ for the four task parameters are $(0.35, 0.25), (0.45, 0.275), (0.40, 0.35), (0.55, 0.400)$. These task parameters are picked within the robot's workspace.

**Obstacle Avoidance Algorithm:** Given a task parameter $\tau$ and an obstacle, the obstacle avoidance task is performed as follows: (i) we sample $z$ from $p(z)$, (ii) generate the bottle's trajectories via $f(z, \tau)$, (iii) check the collision between the bottle and the obstacle and pick collision-free trajectories, and (iv) solve the inverse kinematics problem of the robot and choose one that is feasible and also collision-free.

We check collisions between the bottle and the obstacle and between the robot and the obstacle by converting the meshes of the bottle and robot to point clouds, and parameterizing the obstacle as a superquadric, which represents objects as a sign distance function [57]. As a sign distance function, superquadrics have benefits in checking if a point is inside or outside them. We consider a trajectory of a point cloud and a superquadric to be collision-free if none of the points in the point cloud gets inside the superquadric at every timestep, and consider they collide otherwise.

### D.3.3 Additional results

**Water-Pouring Performance Comparison:** The motions of the bottle pouring water near the cup are highly dependent on the amount of water in the bottle. A bottle of small water needs to be tilted more than a bottle that is almost full to pour the same amount of water into the cup. The amount of tilting of the bottle can be captured in the angle between its axis and the table, which we denote as the bottle angle.

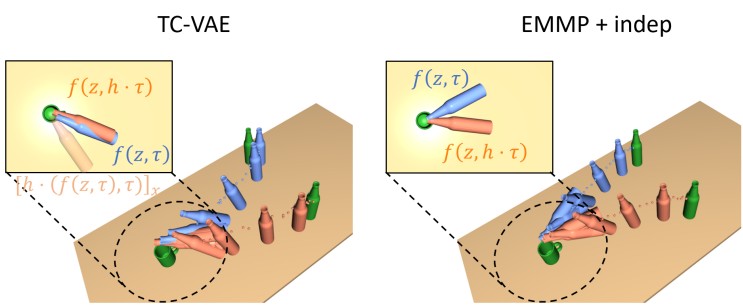

Figure 18: $f(z, \tau)$ (blue) and $f(z, h \cdot \tau)$ (orange), and $[h \cdot (f(z, \tau), \tau)]_x$ (apricot). $[h \cdot (f(z, \tau), \tau)]_x$ and $f(z, h \cdot \tau)$ should be overlapped if the trajectories are generated equivariantly with the task parameters.

Figure 17 illustrates bottle angle mean and standard deviation graphs of demonstration trajectories of the training dataset (*Left*), generated trajectories of EMMP + indep (*Middle*) and generated trajectories of TC-VAE (*Right*) with $m_w = 200$g (blue) and $m_w = 410$g (orange). We randomly augment 50 task parameters of validation and test datasets 20 times, and pick 1,000 task parameters for $m_w = 200$g and 1,000 task parameters for $m_w = 410$g. We generate 1,000 trajectories for both cases using $z$ sampled from $p(z)$.

Figure 17 *Left* shows that as the mass of water increases, the pouring angle increases, which means the bottle is tilted less. It can be seen that the minimum mean angles of EMMP for $m_w = 200$g and $m_w = 410$g (-1.6 degrees and -9.5 degrees) are very much alike that of the demonstration trajectories (-1.5 degrees and -8.5 degrees). On the other hand, the minimum mean angles of TC-VAE (5.6 degrees and -3.1 degrees) are very much distant from the demonstration trajectories'.

**Equivariance Comparison:** For a motion manifold primitive framework to be equivariant, decoded trajectories must equivariantly transform as task parameters undergo a symmetry transformation. We qualitatively compare the equivariance performance of random data augmentation method and equivariant learning method by comparing TC-VAE and EMMP + indep.

Figure 15 *Left* visualizes two trajectories generated from $\tau$ and $h \cdot \tau$, where $h$ is the rotation of the bottle around the cup and itself, without translation. If the model is equivariant, the orange-colored bottle and the apricot-colored bottle in the left upper corner should overlap. However, the condition is not satisfied for TC-VAE, whereas the orange trajectory of EMMP is equivariantly transformed with $\tau$.

