# OpenReview forum: "Equivariant Motion Manifold Primitives"
_robot-learning.org/CoRL/2023/Conference — CoRL 2023 Poster_

### Official Review · Reviewer_ZGAZ · 2023-07-18

**Confidence:** 3
**Originality:** Very Good
**Technical Quality:** Excellent
**Clarity Of Presentation:** Very Good
**Impact:** 4

**Recommendation:**

Weak Accept: I recommend accepting the paper, but will not argue for my recommendation if the majority of other reviewers have a different opinion.

**Review:**

This paper makes an interesting contribution to the field of Learning-by-Demonstration. It introduces a manifold learning technique that enables the learning of the entire space of trajectories required to successfully execute a task.

The paper is commendable for its clear illustrations and explanations. It also includes comprehensive experiments that effectively showcase the technique and demonstrate how to leverage the task symmetries using the proposed framework.

However, to ensure acceptance, a few minor issues need to be addressed.


**Quality Of The Limitations Section:**

Limitations are addressed clearly

**Questions For Rebuttal:**

- In the first paragraph of p. 4, I think it makes more sense to define the symmetry transformation on a generic manifold rather than on $\mathcal{T}$,  since the same notation (i.e. $\cdot$) is used to denote the group action of $H$ on $\mathcal{T}$ and on $\mathcal{X} \times \mathcal{T}$.

- As the paper incorporates abstract algebraic definitions, the term "module" used for $\bar{h}$ might cause confusion initially, as there exists an algebraic structure referred to as a module. To avoid potential confusion, it would be advisable to consider replacing the term "module" with a different term to ensure clarity and prevent any misunderstanding.

- Some titles of the referenced papers should be correctly capitalized, e.g. "Spherical cnns" should be "Spherical CNNs".

- Typos:
    - "equivaraint" at p. 4
    - In B.1. the text "one standard way @ @" seems to be forgotten to be erased


**Robotics Focus:**

Sufficient demonstration on hardware

**Summary Of Paper:**

This paper proposes to use a manifold learning technique to learn manifolds representing the set of all feasible trajectories that successfully execute a task (given a parameter task $\tau$).

Learning the task for all parameters $\tau$ can be very data-intensive. Still, the authors propose some techniques to increase data efficiency and show that their technique is feasible whenever the manifolds (parametrized by $\tau$) are homeomorphic to each other.

To learn such manifolds, the paper proposes an auto-encoder architecture that possesses an invariant encoder and an equivariant decoder. Those take into account the symmetries of a task, and this leads to improved generalization and efficiency in learning tasks.


**Summary Of Recommendation:**

The reviewer finds the paper interesting, with clear presentation and high impact in robotics and machine learning.

---

> ### Author Response · Authors · 2023-08-13
> **Response letter for reviewer ZGAZ**
>
> **COMMON**
>
> Thank you very much for your constructive feedback. In response to the many constructive suggestions we have received, we have spent the past week revising our manuscript accordingly.
> In an attempt to answer the reviewers' questions, and to better clarify and validate our contributions, we include the new additional contents in the revised manuscript as follows:
>
> * in the introduction, we have changed the order between the explanations of independence and equivariance since the equivariance is our main contribution and the equivariant motion manifold does not dependent on the independence;
> * we have added Figure 1 in the introduction that visualizes the trajectory manifold and equivariance;
> * we have changed Figure 2 in Section 2 to clearly illustrate the motion manifold primitives framework;
> * we have changed Figure 3 in Section 3 to clearly illustrate the equivariant motion manifold primitives framework;
> * we have conducted a re-experiment for the water pouring task, securing the bottle to the gripper more tightly;
> * we have added Appendix C (Geometric Preliminaries) for readers not familiar with manifold learning and equivalent models;
> * we have changed the term "equivariant module" to "equivariant map" to reduce confusion;
> * we have added an example of constructing the equivariant map $\bar{h}$ to help understand it;
> * we have fixed some typos.
>
> Below we provide detailed responses to each of the reviewer's comments. When referencing any major changes and the addition of new content to the revised manuscript, we have indicated those passages like `this`.
>
>
> **Q1.** In the first paragraph of p. 4, I think it makes more sense to define the symmetry transformation on a generic manifold rather than on ${\cal T}$, since the same notation (i.e. $\cdot$) is used to denote the group action of $H$ on ${\cal T}$ and on ${\cal X} \times {\cal T}$.
>
> **A1.** Thank you for your suggestion. We agree that the way it was written is confusing since we define the group action separately. Although it would be a great solution to first define the group actions on a generic manifold, since the group actions are defined only on ${\cal T}$ and ${\cal X} \times {\cal T}$, `we have updated the manuscript to first clearly mention that the group actions are defined on the two manifolds.`
>
> **Q2.** As the paper incorporates abstract algebraic definitions, the term "module" used for $\bar{h}$ might cause confusion initially, as there exists an algebraic structure referred to as a module. To avoid potential confusion, it would be advisable to consider replacing the term "module" with a different term to ensure clarity and prevent any misunderstanding.
>
> **A2.** Thank you for raising the point. We agree with you that the term "module" can be confusing. Therefore, we changed the term "equivariant module" to "equivariant map".
>
> **Q3.** Some titles of the referenced papers should be correctly capitalized, e.g. "Spherical cnns" should be "Spherical CNNs".
>
> **A3.** Thank you very much for pointing this out. `We have fixed the title of Spherical CNNs and reviewed the titles once more.`
>
> **Q4.** How do the inductive biases of symmetry+homeomorphism affect sample complexity compared to other approaches?**
>
> **A4.** Thank you for the information. `We have reviewed the paper, and fixed typos including the ones you informed us about.`

---

### Official Review · Reviewer_rfrA · 2023-07-19

**Confidence:** 4
**Originality:** Very Good
**Technical Quality:** Very Good
**Clarity Of Presentation:** Very Good
**Impact:** 4

**Recommendation:**

Strong Accept: I recommend accepting the paper and will argue for my recommendation even if other reviewers hold a different opinion.

**Review:**

Unfortunately, I found it challenging to comprehend the technical aspects of the paper, even though I believe I possess adequate knowledge in robot learning and movement primitives. My reading stopped at Line 93 on page 3 because I became completely lost. The primary reasons for my inability to continue are as follows:

The paper heavily relies on numerous assumptions without providing compelling justifications for their reasonability. Throughout the paper, the phrases "We assume" and "XXX is assumed to be" appear approximately 20 times, particularly on page 2. However, it remains unclear to me why these assumptions are necessary, making it difficult to assess the correctness of the proposed techniques. For instance, I came across an example from Line 53 to Line 56 that discusses the symmetry transformation, where the reasoning behind such an assumption was unclear.

The mathematical descriptions appear disconnected and disjointed, resembling a collection of random sentences. In particular, the lack of coherence between sentences from Line 88 to Line 93 makes it challenging to establish the causality between assumptions and the corresponding mathematical inferences.

To enhance the paper's clarity and credibility, I suggest addressing these issues and providing clearer justifications for the underlying assumptions, as well as improving the flow and coherence of the mathematical descriptions.


$\textbf{Updated review after rebuttal}$:

I appreciate the effort of the authors to improve the readability of the paper. After reading the revised version, I see that my main concern has been addressed. I'm in favor of this paper now.

In the first version, I struggled to understand the theory because of missing explanations and assumed prior knowledge. But in this revised paper:

- The authors added an introduction to the Geometric Preliminaries, which helped me grasp the main theory.
- They've rewritten the methods section and updated the figures, offering a clearer explanation of the method.
- They removed some unnecessary assumptions, making the theory much sounder.

With these changes, I can now see the paper's novelty and contribution. The theory is well-presented now, even for readers without much knowledge in the trajectory manifold. I think the method can be applied to many robot learning tasks. Despite my strong rejection earlier, I'm now leaning towards a strong accept.

**Quality Of The Limitations Section:**

Limitations are addressed clearly

**Questions For Rebuttal:**

I would like to offer some suggestions to the author for improving the paper:

Reevaluate the assumptions: It would be beneficial to reconsider the number of assumptions made in the paper. Aim to retain only the most essential ones while ensuring that they are reasonable and comprehensible to the readers. Over-reliance on assumptions without proper justification can hinder the paper's credibility.

Enhance technical and mathematical descriptions: Provide clearer and more intuitive explanations of the technical and mathematical aspects of the paper. Consider the knowledge level of the readers and avoid assuming that they have the same level of familiarity with the subject matter as the authors. A well-explained and accessible presentation will make it easier for readers to follow and understand the content.

Establish logical connections between sentences: Ensure that the sentences and sections in the paper are logically connected. Show how certain conclusions can be reached based on specific conditions. This will strengthen the coherence and flow of the paper, making it easier for readers to grasp the ideas and arguments presented.

Seek feedback from non-experts: Prior to submitting the manuscript, consider involving individuals with less knowledge in the field to conduct a proofreading and provide feedback. This will help identify areas that may be unclear or challenging for readers who are not experts in the subject matter. Incorporating their feedback can greatly enhance the paper's overall clarity and accessibility.

By considering these improvements, the author can create a more polished and reader-friendly paper that effectively communicates the research findings to a broader audience.


$\textbf{Updated review after rebuttal}$:
My major concern has been addressed and the paper is very friendly to the readers in a broad research area.

**Robotics Focus:**

Sufficient demonstration on hardware

**Summary Of Paper:**

This paper focues on modeling multiple solution trajectories for one task, using movement pritimives and latent manifold.


$\textbf{Updated review after rebuttal}$:

The paper proposed a novel method to learn the trajectory manifold by using the equivariant nature of movements. By constructing the invariant encoder and equivariant decoder strucures, this proposed method is able to capture the equivariant motion manifold primitives. During the rebuttal, the authors have significantly improved the readbility of the paper, makeing it smooth to read and technically sound.

**Summary Of Recommendation:**

I regret to express my reservation about recommending this paper as I encountered considerable difficulty comprehending the technical details. The abundance of assumptions presented throughout the paper lacks adequate and reasonable explanations, contributing to the overall confusion in the writing.


$\textbf{Updated review after rebuttal}$:

I recommend this paper now, as the revised version has improved the readability a lot. After breaking through the bottleneck of the technical part, I found the idea of the paper is very interesting and should be capable to generalize to a lot of similar tasks.

---

> ### Author Response · Authors · 2023-08-13
> **Response letter for reviewer rfrA**
>
> We also made more changes to improve the readability of the paper in response to the concerns of the reviewers as follows: `(i) we have revised the first paragraph of Section 2.1 (Line 84 - Line 92) to have a better flow and be more coherent, (ii) have added an example of constructing the equivariant map $\bar{h}$ to help the reader understand, (iii) have changed Figure 2 (which illustrates the concept of MMP) so that our concept could be understood better, and (iv) have changed Figure 3 (which illustrated the concept of EMMP) so that our concept could be understood better. To further help readers comprehend the paper, we added Appendix C (Geometric Preliminaries) which consists of prior knowledge to understand manifold hypothesis, homeomorphism, group, and equivariance, and we mentioned it at the end of the first paragraph of Section 2 and Section 3.`
>
> About the over usage of the term "assume", we have removed, `(i) the assumption that the manifold can have multiple connected components (Line 79) and (ii) the assumption that the support of $p(z|\tau)$ is a compact subspace of $\mathbb{R}^m$ (Line 91), which we found are not strictly necessary to apply our framework.`
>
> For the remaining assumptions, we assure you that they are reasonable, not very strict, and hold true for examples in our experiments (and likely to be true for many other examples as well). First, the assumption that the high-dimensional demonstration trajectories form a low-dimensional manifold -- which is known as the manifold hypothesis in the machine learning community and has been widely adopted -- is not strict since it is impossible for the set of finite (often a few) trajectories to fill up the entire trajectory data space (as visualized in the newly added Appendix C.1). It is reasonable to assume that there is a lower-dimensional space where the demonstration trajectory data approximately lie, although choosing a proper latent space dimension (or manifold dimension) can indeed be a challenging task.
>
> Second, regarding the homeomorphic manifold assumption -- where we assume that the motion manifold primitives for each task parameter can be deformed smoothly into each other --, we want to first point out that our equivariant manifold primitives can be used even without this assumption (please see section 3.2, where we first define an encoder without the assumption and then introduce the simplified version with the assumption). But we did take this assumption in our experiments to improve the sample efficiency because the assumption holds in our examples.
>
> Third, the equivariant manifold primitive assumption -- where we assume that the motion manifold should be transformed equivariantly under the symmetry transformation on the task parameter -- is the key assumption that we believe many problems in robotics satisfy, including the examples shown in the paper. One of the reasons is that many physical objects and environments the robot needs to interact with often have symmetries such as the cylindrical symmetry of the bottle and cup.  We believe capturing this symmetry is one of the important features humans have when perceiving and interacting with environments. We have added an illustrative figure in the introduction section to describe this more clearly.
>
> Lastly, we made the assumption that the manifold can be parametrized using a single non-linear mapping $f:\mathbb{R}^{m} \to \mathbb{R}^{D}$ in order to use the standard autoencoder-based manifold learning methods, which we admit as a limitation of our current framework. When the manifold cannot have a single global coordinate chart (e.g., $S^1 \subset \mathbb{R}^{3}$ needs at least two local coordinate charts), using a single coordinate system $f:\mathbb{R}^{1} \to \mathbb{R}^{3}$ cannot lead to a complete parametrization of the manifold $S^1$ (due to topological difference). In our current framework, we can practically set the latent space dimension to be 2 -- although the manifold dimension is 1 --, and then make the latent density model $p(z)$ find $S^1$ in the latent space $\mathbb{R}^{2}$. In future research, we believe using multiple mappings $f$ could be a better way, and our key idea of using the invariant encoder and equivariant decoder to learn equivariant manifold primitives can be naturally extended.

---

> ### Author Response · Authors · 2023-08-13
> **Response letter for reviewer rfrA**
>
> **COMMON**
>
> Thank you very much for your constructive feedback. In response to the many constructive suggestions we have received, we have spent the past week revising our manuscript accordingly.
> In an attempt to answer the reviewers' questions, and to better clarify and validate our contributions, we include the new additional contents in the revised manuscript as follows:
>
> * in the introduction, we have changed the order between the explanations of independence and equivariance since the equivariance is our main contribution and the equivariant motion manifold does not dependent on the independence;
> * we have added Figure 1 in the introduction that visualizes the trajectory manifold and equivariance;
> * we have changed Figure 2 in Section 2 to clearly illustrate the motion manifold primitives framework;
> * we have changed Figure 3 in Section 3 to clearly illustrate the equivariant motion manifold primitives framework;
> * we have conducted a re-experiment for the water pouring task, securing the bottle to the gripper more tightly;
> * we have added Appendix C (Geometric Preliminaries) for readers not familiar with manifold learning and equivalent models;
> * we have changed the term "equivariant module" to "equivariant map" to reduce confusion;
> * we have added an example of constructing the equivariant map $\bar{h}$ to help understand it;
> * we have fixed some typos.
>
> Below we provide detailed responses to each of the reviewer's comments. When referencing any major changes and the addition of new content to the revised manuscript, we have indicated those passages like `this`.
>
>
> We greatly appreciate your sincere concerns. We have reviewed the paper again and realized that the paper can be difficult, especially for readers who are not familiar with the concepts of *manifold* and *equivariant models*, and that we may have overused the term "assume" even when it is not strictly necessary. For the past week, we have taken steps to amend the manuscript on those two points.
>
> Before going into the details, although it may already be apparent to you, we would like to first emphasize that our main contribution is the construction of the *equivariant* manifold primitives. Learning a continuous manifold of motion rajectories -- which we call the manifold primitives -- by using the autoencoder framework is first done by the recent work TC-VAE [37], however, this approach shows less-than-desirable performance given a small dataset. In this work, we have shown that the equivariant manifold primitives can greatly reduce the complexity of the learning problem and significantly improve performance.
>
> `However, we have realized that "what is equivariance" and "why the motion manifold primitives should be equivariant" may not be sufficiently well described in the introduction section. Thus we have added an illustrative figure (Figure 1) that visualizes the concept of equivariance, and revised the section accordingly.`

---

### Official Review · Reviewer_K1vn · 2023-07-19

**Confidence:** 3
**Originality:** Very Good
**Technical Quality:** Excellent
**Clarity Of Presentation:** Excellent
**Impact:** 4

**Recommendation:**

Strong Accept: I recommend accepting the paper and will argue for my recommendation even if other reviewers hold a different opinion.

**Review:**

The paper is well written, the derivations are thorough, and the proposed method is well supported with simulation and real-world results. Ample comparison and ablation of the methods are provided.

The appendix and supplementary video provide detailed information about the submission and are both helpful for understanding the paper.

Please refer to the “questions for rebuttal” section for more details.

**Quality Of The Limitations Section:**

Limitations are addressed clearly

**Questions For Rebuttal:**

1.	The trajectories illustrated in figure 1 do not belong to the same homotopy class. Doesn’t this break the assumption that $\mathcal{M}_{\tau}$ is homeomorphic?

2.	I think the architecture comparison in C.2.3 are quite interesting and should be included in the main submission if possible.

3.	The speculation in section 4.2 on “slipping between the two-finger gripper and the bottle” seems to be something easily fixable by attaching the bottle rigidly to the end effector. I would be very interested in the performance of the method once the slippage is fixed.

4.	Typo: the definition of “SR” is omitted in Table 2. Is it “success rate”?

**Robotics Focus:**

Sufficient demonstration on hardware

**Summary Of Paper:**

This paper proposes learning a continuous manifold of trajectories, referred to as Motion Manifold Primitives (MMP), to achieve a specified task. Compared to single trajectories, MMP is able to address unforeseen constraints and obstacles. The authors further propose EMMP, which additionally accounts for equivariant transformations through using an invariant encoder and an equivariant decoder. The proposed methods were applied to synthetic and real-world robot experiments and compared against exiting manifold primitive methods, with EMMP showing superior results. The authors demonstrated that the invariant encoder and equivariant decoder are key factors for performance improvement through ablations and swapping different network architectures.

**Summary Of Recommendation:**

This paper presents a method that is rigorously justified, the results are favorable, and the comparisons are comprehensive. I recommend accepting this paper to CoRL 2023.

---

> ### Author Response · Authors · 2023-08-13
> **Response letter for reviewer K1vn**
>
> **COMMON**
>
> Thank you very much for your constructive feedback. In response to the many constructive suggestions we have received, we have spent the past week revising our manuscript accordingly.
> In an attempt to answer the reviewers' questions, and to better clarify and validate our contributions, we include the new additional contents in the revised manuscript as follows:
>
> * in the introduction, we have changed the order between the explanations of independence and equivariance since the equivariance is our main contribution and the equivariant motion manifold does not dependent on the independence;
> * we have added Figure 1 in the introduction that visualizes the trajectory manifold and equivariance;
> * we have changed Figure 2 in Section 2 to clearly illustrate the motion manifold primitives framework;
> * we have changed Figure 3 in Section 3 to clearly illustrate the equivariant motion manifold primitives framework;
> * we have conducted a re-experiment for the water pouring task, securing the bottle to the gripper more tightly;
> * we have added Appendix C (Geometric Preliminaries) for readers not familiar with manifold learning and equivalent models;
> * we have changed the term "equivariant module" to "equivariant map" to reduce confusion;
> * we have added an example of constructing the equivariant map $\bar{h}$ to help understand it;
> * we have fixed some typos.
>
> Below we provide detailed responses to each of the reviewer's comments. When referencing any major changes and the addition of new content to the revised manuscript, we have indicated those passages like `this`
>
> **Q1.** The trajectories illustrated in Figure 1 do not belong to the same homotopy class. Doesn’t this break the assumption that ${\cal M}_\tau$ is homeomorphic?
>
> **A1.** The homeomorphic manifold assumption does not assume that the trajectories on ${\cal M}_\tau$ are homeomorphic, but rather assumes that multiple ${\cal M}_\tau$'s are homeomorphic to each other. The ${\cal M}_\tau$ of Figure 1 contains trajectories of two homotopy classes. In this case, as long as other ${\cal M}_\tau$'s also contain trajectories of the two homotopy classes as Figure 1 does, the homeomorphic assumption holds.
>
> **Q2.** I think the architecture comparison in C.2.3 are quite interesting and should be included in the main submission if possible.
>
> **A2.** Thank you for your suggestion. I agree with you that the architecture comparison needs to be in the experiment section. Unfortunately, because of the CoRL's 8-page limitation, we could not include the architecture comparison in the main manuscript.
>
> **Q3.** The speculation in section 4.2 on “slipping between the two-finger gripper and the bottle” seems to be something easily fixable by attaching the bottle rigidly to the end effector. I would be very interested in the performance of the method once the slippage is fixed.
>
> **A3.**
> Thank you for raising this point. We have found a way to fix the bottle to the end-effector tightly. `We have added the bottle-fixing
> process in the supplementary video. We have redone the water-pouring experiment and updated the new result in Table 2 in the paper.` As a result, the performance of our model has improved significantly, while the performance of the baseline TC-VAE has not. Specifically, the average water pouring error of our model has reduced from 37g to 23g, that of the baseline model remained at 87g, and that of the replayed demonstration has reduced from 25g to 19g.

---

### Official Review · Reviewer_3SRv · 2023-07-21

**Confidence:** 5
**Originality:** Very Good
**Technical Quality:** Very Good
**Clarity Of Presentation:** Good
**Impact:** 3

**Recommendation:**

Strong Accept: I recommend accepting the paper and will argue for my recommendation even if other reviewers hold a different opinion.

**Review:**

# Originality

Manifold learning in and of itself is not novel - mapping a density function on a lower-dimensional manifold which is easy to sample from to a high-dimensional manifold is the backbone of most autoencoding methods. So learning a mapping between these manifolds for robotics tasks is not novel.

However, this work attempts to drill down into the structure of these manifolds in many real robotics tasks, and make some more precise guarantees on performance by exploiting this structure. Broadly speaking, this work is attempting to formally nail down some of the properties which are reasonably common in robotics tasks:
Problem symmetries: the solutions set to motion planning problems, for instance, is often equivariant under some certain transformations of initial conditions. We can often enumerate these symmetries intuitively, but the community frequently struggles to deal with symmetries systematically (often offloading them to multimodal generative modeling frameworks).
Homeomorphism: If you change the task parameters slightly, you’ll probably be able to deform the solution manifolds smoothly as well. Intuitively, if you move an object back a little bit during a pick-and-place task, your manifold of pick-and-place motions will probably deform accordingly, and not change topologically. Some methods exhibit these properties (for instance, DMPs), but not necessarily when learning manifolds.

These observations are not in and of themselves novel - lots of work has noticed these things. But very little work has tried to tackle them explicitly and from first principles, and design solutions which exploit the mathematical structure of these properties. And I haven’t really seen much / any work which tries to rope in group theory / formal symmetry properties into trajectory learning. So to me, this is quite original for robot learning - I learned something by reading the paper, and I think the community would benefit from thinking about these properties with a more rigorous lens.

# Quality

## Theoretical

As far as I can tell, the theoretical contributions of this work are sound.

One question: is the independence regularization term from prior work? Or designed for this work?

## Empirical

The two examples are reasonably well-executed. Both the toy task and the real-world robot task have rigorously-defined symmetries, and

However, I guess I have the following issue with the baselines: if a task is simple enough to write down all the precise symmetries for, then sufficiently rich neural network models and a sufficiently large set of demonstrations should be able to represent these symmetries as well. More recent work than TC-VAE has shown impressive ability to represent multimodal manifolds (i.e. [1]), so I’d be curious how much of a benefit leveraging these simple explicit symmetries actually has over more recent techniques.

[1] Diffusion Policy: Visuomotor Policy Learning via Action Diffusion

# Clarity

My biggest issue with the paper is that the mathematical treatment of symmetry was a bit rushed + inaccessible. For this community, it’s unlikely that people will be familiar with the differences between group operations and group actions (and may conflate the two), when in fact that distinction is actually quite important to understanding what the symmetry operators are doing. The symmetry operators should at the very least have a larger diagram which relates the notation to an actual example. Figure 2 should be improved, made larger, and related to the operators better. Something with transition arrows like found on [1]. Also, section 3.2 doesn’t have much intuition, and it’s really not clear intuitively how h^bar can be constructed or is invertible. It would be really, really useful to incorporate the toy example here while explaining it.

There isn’t really enough of a related work section. If it doesn’t fit in the main paper, it should be put in the appendix and referenced in the main paper.

[1] https://en.wikipedia.org/wiki/Symmetry_group

# Significance

Another issue I have is in the significance of the work given the assumptions. I’m not as concerned with the homeomorphism assumption as I am with the assumption that mathematically enumerating the symmetries in arbitrary robot learning problems is simple or even possible. In most real-world problems, there are only rough/fuzzy/approximate symmetries that can be exploited, and it would very difficult to write down these symmetries, let alone define an h^bar which can be used to construct an equivariant function under those symmetries. Furthermore, if you’re trying to solve a CLASS of tasks with different symmetries, it’s unclear how you’de represent that in this framework without explicitly enumerating each element in that class. For instance, consider hanging a mug on a rack - there are discrete and rotational symmetries which may be different for different mugs and racks, particularly if the geometries are rather different.

# Relevance

The work is undeniably relevant to robot learning literature - we deal with symmetries very frequently, and many works fail to address them at all, let alone with mathematical rigor. I think many people will find this treatment very interesting + inspiring in this setting.

# Limitations

The authors do an adequate job addressing the two assumptions (homeomorphism, ease of defining h^bar). However, they don’t really mention how the method might be used in a situation where you have a set of symmetry groups you might want one method to solve. This puts a pretty hard cap on the direct utility of this method in real-world learning scenarios.

Not sure if this is a limitation per se, but the homeomorphic assumption is not particularly elegant when it comes to predicting trajectories which are time-parametereized. For instance, stretching a short trajectory to a long trajectory with the (presumably) same set of time-indexed waypoints doesn’t really make so much sense for control. Instead, would be interesting to see if the method could be adapted to some kind of time-invariant trajectory shape through configuration space.

Finally, the method doesn’t really make any discussion of manifold boundaries, which are quite relevant for many real robot tasks.


**Quality Of The Limitations Section:**

Limitations are addressed clearly

**Questions For Rebuttal:**

How do the inductive biases of symmetry+homeomorphism affect sample complexity compared to other approaches?

Is the independence regularization term from prior work? Or designed for this work?

How would the authors extend their work to classes of tasks where each instance of hte task may have different symmetries?


(Not really asking for this experiment in rebuttal, but I am interested in some discussion): How would this method compare to diffusion-based approaches, such as Diffusion Policy, which have been shown empirically to represent configuration-space manifolds quite accurately?


**Robotics Focus:**

Sufficient demonstration on hardware

**Summary Of Paper:**

In this work, the authors address the task of learning sets of valid configuration-space trajectories for arbitrary control tasks. The authors propose Motion Manifold Primitives, which is a mapping from a simpler low-dimensional space (i.e. R^m) to the configuration space. The core idea is that one can learn a density function over the latent space which can then be sampled to recover samples from the manifold in the configuration space. The authors’ second contribution is noticing that if the task parameter space has symmetries, and those symmetries can be cleanly and precisely described, and an equivariant group action can be defined, then one can formulate a mapping from the latent space to configuration space which is equivariant under all enumerated symmetries. The authors provide convincing empirical evidence that, in two scenarios with straightforward symmetries, their method demonstrates superior performance compared to other manifold-learning baselines.


**Summary Of Recommendation:**

This paper offers a compelling formal perspective on handling symmetry in robot learning tasks. While I'm not entirely sure about the applicability of this method to a broader range of more-complicated robot problems because it is difficult to exactly satisfy the assumptions, I certainly learned something by reading this paper and believe that the CoRL community would benefit from considering the implications of the theoretical treatments of symmetry/equivariance.

---

> ### Author Response · Authors · 2023-08-13
> **Response letter for reviewer 3SRv**
>
>
> **Q1.** How do the inductive biases of symmetry+homeomorphism affect sample complexity compared to other approaches?
>
> **A1.** Those biases reduce the input and output space dimensions of the functions that we have to learn such as the decoder and latent density function, and the sample complexity decreases accordingly.
>
> First, in autoencoder-based manifold learning, we fit a decoder $f:\mathbb{R}^{m} \times {\cal T} \to {\cal X}$. Given the task parameter space dimension $d_{\tau}$, the input space dimension is $m + d_{\tau}$ in the existing method. With our equivariance inductive bias with a symmetry group $H$, we now only need to learn $f$ in the quotient space $(\mathbb{R}^{m} \times {\cal T})/H$ that is much smaller than the original input space. This significantly reduces the sample complexity.
>
> Second, in latent density learning, we need to fit $p(z|\tau)$ without the homeomorphic manifold assumption. In this case, similarly, while the existing method needs to learn $p(z|\tau)$ in the entire task space ${\cal T}$, our method needs to learn it in the quotient space ${\cal T}/H$. This also decreases the sample complexity.
>
> Third, suppose we take the homeomorphic manifold assumption. Then we only need to learn $p(z)$ instead of $p(z|\tau)$, meaning that the input space dimension has been decreased by the dimension of ${\cal T}$. Therefore, the sample complexity decreases. In this case, additionally, we need to make $z$ and $\tau$ independent, yet enforcing the independence between $z$ and $\tau$ may not be a simple task. The symmetry bias takes an additional role here, where it reduces ${\cal T}$ to ${\cal T}/H$ and reduces the sample complexity of the task of making $z$ and $\tau$ independent.
>
> **Q2.** Is the independence regularization term from prior work? Or designed for this work?
>
> **A2.** We have newly designed the independence regularization term for this work. `To emphasize this fact, we added "newly" on page 2, line 53 of the paper.`
>
> **Q3.** How do the inductive biases of symmetry+homeomorphism affect sample complexity compared to other approaches?
>
> **A3.** Thank you for the great question. Given multiple different symmetry groups, one possible approach is to define a direct product group and corresponding group action to ${\cal T}$. Then, we can apply our framework with a suitable design of the equivariant map $\bar{h}$ without changes in our framework.
>
> There may exist some conditions for the product group action to be well-defined, and in such cases, identifying the sufficient and necessary conditions for the groups and actions would be an important issue. In addition, the construction of the equivariant map $\bar{h}$ may pose another challenge. It would be interesting to develop a framework that combines a set of pre-defined maps $\bar{h}$ for each group to construct a valid $\bar{h}$ for the product group.
>
> **Q4.** How do the inductive biases of symmetry+homeomorphism affect sample complexity compared to other approaches?
>
> **A4.** In general, diffusion models are not biased to learn a low-dimensional manifold structure. Since the noise space dimension is the same as the data space dimension, if the underlying data manifold dimension is much smaller than the data space dimension, the diffusion process can struggle to accurately learn the manifold structure. In such cases, we believe combining autoencoder-based manifold learning and a latent space diffusion model for identifying boundaries can be very powerful, as shown in the large-scale image generative model that uses the latent diffusion models.
>
> **Q5.** How do the inductive biases of symmetry+homeomorphism affect sample complexity compared to other approaches?
>
> **A5.** Thank you very much for raising this issue. `Taking your precious advice, we have newly added Figure 1 to illustrate the motion manifold primitives and equivariance.We also have changed Figure 2 and Figure 3 tohelp understand the concept of MMP and EMMP respectively. Additionally, we have added an example of how we construct the $\bar{h}$. Finally, we have added a geometric preliminary section in the appendix to help people understand.`

---

> > ### Author Response · Authors · 2023-08-13
> > **Response letter for reviewer 3SRv**
> >
> > **COMMON**
> >
> > Thank you very much for your constructive feedback. In response to the many constructive suggestions we have received, we have spent the past week revising our manuscript accordingly.
> > In an attempt to answer the reviewers' questions, and to better clarify and validate our contributions, we include the new additional contents in the revised manuscript as follows:
> >
> > * in the introduction, we have changed the order between the explanations of independence and equivariance since the equivariance is our main contribution and the equivariant motion manifold does not dependent on the independence;
> > * we have added Figure 1 in the introduction that visualizes the trajectory manifold and equivariance;
> > * we have changed Figure 2 in Section 2 to clearly illustrate the motion manifold primitives framework;
> > * we have changed Figure 3 in Section 3 to clearly illustrate the equivariant motion manifold primitives framework;
> > * we have conducted a re-experiment for the water pouring task, securing the bottle to the gripper more tightly;
> > * we have added Appendix C (Geometric Preliminaries) for readers not familiar with manifold learning and equivalent models;
> > * we have changed the term "equivariant module" to "equivariant map" to reduce confusion;
> > * we have added an example of constructing the equivariant map $\bar{h}$ to help understand it;
> > * we have fixed some typos.
> >
> > Below we provide detailed responses to each of the reviewer's comments. When referencing any major changes and the addition of new content to the revised manuscript, we have indicated those passages like `this`.

---

### Decision · Program_Chairs · 2023-08-30

**Decision:**

Accept (Poster)

**Comment:**

The paper presents an imitation learning framework to model a continuous manifold of demonstrated trajectories. To address the issue of the limited data, the proposed method leverages symmetries in robot tasks.
While issues of the presentation in the paper were raised in the initial review, the presentation was significantly improved after the rebuttal.
As the reviewers agreed to accept the paper, AE recommends the acceptance of the paper.